# Copy number variation introduced by a massive mobile element facilitates global thermal adaptation in a fungal wheat pathogen

Sabina Moser Tralamazza[1], Emile Gluck-Thaler [1,3], Alice Feurtey[1,2] & Daniel Croll [1] ✉

Copy number variation (CNV) can drive rapid evolution in changing environments. In microbial pathogens, such adaptation is a key factor underpinning epidemics and colonization of new niches. However, the genomic determinants of such adaptation remain poorly understood. Here, we systematically investigate CNVs in a large genome sequencing dataset spanning a worldwide collection of 1104 genomes from the major wheat pathogen *Zymoseptoria tritici*. We found overall strong purifying selection acting on most CNVs. Genomic defense mechanisms likely accelerated gene loss over episodes of continental colonization. Local adaptation along climatic gradients was likely facilitated by CNVs affecting secondary metabolite production and gene loss in general. One of the strongest loci for climatic adaptation is a highly conserved gene of the NAD-dependent Sirtuin family. The *Sirtuin* CNV locus localizes to an ~68-kb *Starship* mobile element unique to the species carrying genes highly expressed during plant infection. The element has likely lost the ability to transpose, demonstrating how the ongoing domestication of cargo-carrying selfish elements can contribute to selectable variation within populations. Our work highlights how standing variation in gene copy numbers at the global scale can be a major factor driving climatic and metabolic adaptation in microbial species.

Populations occupying heterogeneous environments may evolve locally advantageous traits under divergent selection pressures[1]. How different forms of genetic variation contribute to such environmental adaptation remains unclear. Most broad-scale comparative evolutionary analyses are focused on single nucleotide variants (SNVs)[2–4]. However, large-effect structural variants have also been shown to play a role in species range adaptation[5–9]. Adaptative chromosomal inversions are well documented across populations of *Drosophila* and are linked to seasonal temperature fluctuations[6,7,10] and cold tolerance[8,9]. Copy number variation (CNV) is a type of unbalanced structural variant defined by the loss or gain of sequence fragments ranging from ~50 bp in length to entire chromosomes. Analyzing CNVs systematically remains challenging due to limits in the detection and resolution of the exact sequence rearrangements[11,12]. CNVs can drive genome evolution[13], contribute to domestication and speciation events[14,15], and promote environmental adaptation[16,17]. Population-based studies

[1]Laboratory of Evolutionary Genetics, Institute of Biology, University of Neuchâtel, CH-2000 Neuchâtel, Switzerland. [2]Plant Pathology, D-USYS, ETH Zurich, CH-8092 Zurich, Switzerland. [3]Present address: Department of Plant Pathology, University of Wisconsin-Madison, Madison, WI, USA. ✉e-mail: daniel.croll@unine.ch

revealed CNVs associated with environmental adaptation in seabirds, with a large 60 kb CNV likely contributing to plumage and thermal adaptation[18]. In wild lobster populations, CNVs but not SNVs are associated with sea surface temperature adaptation[19]. Hence, elucidating the population genetic context of widely distributed species is necessary to assess how CNVs and dynamic genome compartments contribute to adaptation. The impacts of gene gains and losses mediated by CNVs across the genome vary from local gene dosage effects[20] to reshuffling gene structures[21], global transcriptional changes, and chromatin reconfiguration[22,23]. CNVs mainly arise from inaccurate DNA repair and nonhomologous recombination[24]. Segmental duplications are often triggered by transposable element (TE) activity, and simple repeats are targets for nonallelic homologous recombination (NHR), leading to CNVs[25,26]. Overall, CNVs are linked to replicative or non-replicative nonhomologous processes based on weak homology[24,27].

CNVs have been implicated in numerous phenotypic traits, including human disease[28], life-history traits of crops[29,30], and drug resistance[31,32]. For example, gene duplication of *ACE-1*, the target site for organophosphate and carbamate insecticides, confers resistance to the *malaria vector Anopheles gambiae*[31]. CNVs in fungal plant pathogens are a major concern because such genetic variation is linked to fungicide resistance[33,34], pathogen virulence[35,36], and nutrient absorption efficiency[37]. Rapid adaptation in plant pathogens is a threat to global food security[38] and facilitates climate change[39,40]. *Zymoseptoria tritici* is one of the most destructive pathogens of wheat crops worldwide[41]. This haploid ascomycete underwent global population expansion concurrent with the introduction of wheat cultivation across continents[42]. With global spread, the pathogen accumulated mutations likely involved in adaptation to new climates[42]. The genome is organized in highly dynamic chromosomes, including eight accessory chromosomes and high degrees of structural variation[43,44]. The genome has expanded recently, most likely caused by TE activity and a weakening of genomic defense mediated by repeat-induced point mutations (RIPs)[42,45]. RIP is thought to be active during sexual reproduction and promotes mutations in any duplicated sequences[46]. Hence, the genomic defense mechanism is thought to constrain adaptation through CNVs. The species exhibits high gene set polymorphisms across populations[47]; however, how the global spread of the wheat host has shaped environmental adaptation remains unknown.

Here, we analyze a global panel of 1109 *Z. tritici* genomes covering all major regions linked to the domestication history of the wheat host. We validate a set of high-confidence CNVs to recapitulate the evolution of gene gain and loss across the global population genetic context of the species to assess the impact on gene functions. Finally, we show how CNVs contributed to chromosomal polymorphism and environmental range adaptation, including an -68 kb cargo-carrying *Starship* element.

## Results

### Chromosomal and gene copy number variants in a 1000-genome panel

We used short-read sequencing data generated for a global panel of 1109 genomes covering the global distribution range of *Z. tritici* (Fig. 1A). The collection of genomes covers 42 countries, capturing the spread of the pathogen concurrently with the historic spread of wheat cultivation across continents[42]. The genomes were collected from a broad range of climates from hot and dry Middle Eastern regions to cooler and humid regions at high latitudes. We performed short-read mapping along the genome to assess segmental deletions and duplications. We implemented multiple filtering procedures and validation steps to ensure high-confidence CNV calls (Fig. 1B). To evaluate CNV call performance, we first compared the gene CNV calling of seven matching pair strains with replicated sequencing data. We found largely congruent calls for duplications and deletions (Fig. 1C). Next, we

validated the CNV calling independently based on completely assembled genomes and synteny analyses[48] (Supplementary Data 3). We found high consistency for variant calls between short-read CNV calling and the chromosome-level synteny approach (Fig. 1D). We used the empirically assessed confidence threshold (*i.e.*, CNQ) to filter the global short-read dataset. Deletion and single-copy event calls were filtered to reduce false positives and retain high-confidence calls (Fig. 1D, E, Supplementary Fig. S1A, B. Finally, we evaluated CNV call quality for 14 core chromosome genes based on PCR assays conducted on 18 strains[47]. We compared against the CNV unfiltered dataset and the filtered dataset (Fig. 1F). The resulting dataset of high-confidence calls included 1104 strains and 8625 distinct gene loci affected by CNVs (Fig. 1B; Supplementary Data 4).

The species carries a set of eight highly polymorphic accessory chromosomes with unknown contributions to environmental adaptation. Accessory chromosomes are more polymorphic compared to core chromosomes[44], creating challenges to define clear read depth thresholds for chromosome presence and absence (Fig. 2A). Shorter accessory chromosome variants of the canonical chromosome variant are expected to show reduced overall read depth due to missing segments. We applied a threshold of >60% genes per chromosome showing deletions to call the chromosome missing. Similarly, we required >60% of the genes to be called duplicated to call an accessory chromosome duplication (Fig. 2B, Supplementary Fig. S2A). Based on these thresholds, chromosomes with very substantial segmental deletions were considered in the same category as complete losses. Similarly, substantial partial duplications affecting were grouped with complete duplications (Supplementary Fig. S2B). To assess the reliability of chromosome CNV calling, we assessed the presence of accessory chromosomes in eight strains where a chromosome-level assembly was produced using PacBio long-read sequencing. We found a 100% match between chromosome CNV calling and chromosome sets present in the assemblies (Supplementary Fig. S2C). Furthermore, we confirmed the absence of particular chromosomes using transcriptomics data (Supplementary Fig. S3A)[48]. Finally, we verified accessory chromosome CNV calls using data from a previously conducted PCR assay covering 71 loci across two cores and all eight accessory chromosomes[49]. A total of 59 strains overlapped the global genome panel used in this study and the PCR assay. We defined missing chromosomes based on the PCR essay if 45% or more loci (a total mean of 7 loci tested per chromosome) were missing within the respective chromosome. We could validate 99.5% of chromosome presence calls and 97% of all chromosome absence calls (Supplementary Fig. S3B).

As expected, core chromosomes 1–13 were fixed in the global collection (Fig. 2C). We found 17 (0.1%) cases of core chromosome duplications (full or partial), with chromosomes 5 and 12 accounting for 64% of all cases (Fig. 2D). We found that 19% of accessory chromosomes were missing from the global collection (*n* = 1698 out of 8872; Fig. 2C). Chromosome 18 was missing in 57% of strains, followed by chromosome 21, which was missing in 26% of samples (Fig. 2C). Overall, chromosome 18 accounts for 70% of all complex arrangements (i.e., high degree of duplications and deletions across the chromosome arm; Supplementary Fig. S3C), with most variation being associated with high repeat content. We found that the chromosome 14 structural variation in the population was associated with a large insertion[44,49] of 351 kb encoding ca. 40 transcriptionally repressed genes (Supplementary Fig. S3D). The total chromosome number per genome varied up to 56% (13–23 chromosomes), with an average of 20 chromosomes underpinning substantial genetic diversity (Fig. 2E). Two strains (0.6% of total) carried only core chromosomes (Supplementary Fig. S3E). We found no evidence supporting yet undescribed accessory chromosomes analyzing scaffolds produced by de novo genome assembly. Higher chromosome numbers than found in the reference genome strain were caused by chromosomal duplications.

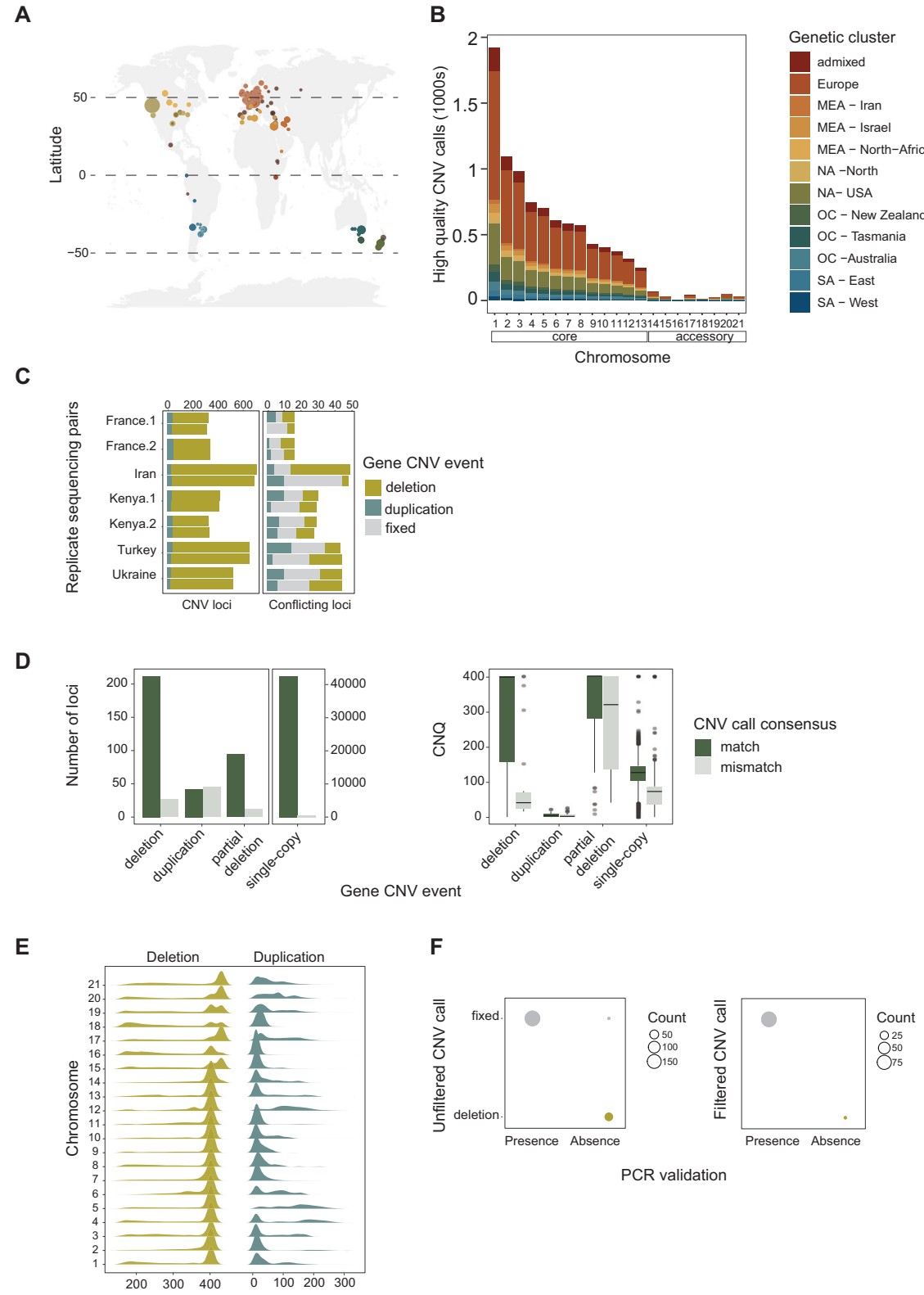

We found that even strains with the highest chromosome numbers ($n \geq 21$) were able to successfully infect wheat leaves and reproduce (Supplementary Fig. S3F).

To identify candidate CNVs associated with climatic adaptation, we analyzed gene CNV variation across the global distribution range of the pathogen. We focused on biallelic CNVs with single-copy, deletion, or duplication of genes (Supplementary Fig. S4A). For each strain, missing or duplicated chromosomes were removed to retain only

single-copy chromosomes for further analyses. We found that 3%, 5.1%, and 18.3% of CNVs found in genes were common (>1% CNV frequency), rare (≤1% CNV frequency), and singletons, respectively, across the 1000-genome panel for a total of 8511 loci (Fig. 3A). Most genes (73.9%) showed no CNVs (i.e. fixed CNV frequency category). Most CNV genes (85.7%) share an ortholog in at least one sister species (Supplementary Fig. S4B). Using parsimony, we infer that most deletions of CNV events are likely gene losses. Most genes on accessory chromosomes (90%;

**Fig. 1 | High-quality gene CNV calls across a thousand-genome panel of the wheat pathogen *Zymoseptoria tritici*. A** Geographic distribution of the samples. N=1109 samples. The world map was created with the R package *maps* v. 3.4.2 based on Natural Earth project data (https://www.naturalearthdata.com). Source data can be found in Supplementary Data 1. **B** CNV call dataset after hard filtering across genetic clusters. Genetic clusters reflect the main genetic differentiation across the globe and were retrieved from Feurtey et al.[42] Source data can be found in Supplementary Data 4. **C** Gene CNV call accuracy comparison between strains sequenced twice. Conflicting loci are defined as mismatches in CNV calls among the sequencing replicates. Source data can be found in Supplementary Data 4. **D** Gene CNV calls congruence comparison between GATK and chromosome-level assembly (SyRI) based CNV calling methods. The bar plot refers to the number of matching loci between methods. *N* = 43,517 loci. The boxplot refers to the quality call parameter CNQ score between matching and nonmatching calls. The box center line represents the median, and the limits represent the first and third quartiles. Whiskers indicate maximum and minimum values. Source data can be found in Supplementary Data 3. **E** CNQ score distribution for duplications and deletions per chromosome after filtering. Source data can be found in Supplementary Data 4. **F** Number of matches of gene CNV calls with PCR validation (14 core chromosome genes tested in 18 strains). The comparison shows both matches with the unfiltered and the filtered CNV callset. Circle size reflects the number of occurrences.

$n = 203$) exhibited CNVs compared to 24% ($n = 2021$) of core chromosome genes (Fig. 3A). Among the core chromosomes, chromosome 5 exhibited the strongest skew toward low-frequency CNVs (*i.e.*, singleton and rare category with CNV frequency ≤1%; Fig. 3A). CNV variation was also found across chromosome arms (Fig. 3B). We found overall gene duplications to be more abundant ($n = 1400$) but mostly at low frequency (1388 CNVs with ≤1% duplication frequency in panel) compared to gene deletions ($n = 682$, Fig. 3C; Supplementary Fig. S4C). The distinct frequency patterns of duplications and deletions were consistent across quality filtering stages (i.e. low and high-frequency CNVs; Supplementary Fig. S4D). Hence, we asked whether gene deletions and duplications would segregate differently among populations (Fig. 3D). Rare CNVs (global frequency <1%) showed similar proportions for gene deletions and duplications compared to the common frequency category (>1%). Gene duplications were four times less likely shared between populations compared to deletions ($p$-value < 0.00001, 95% CI 14−41%; Supplementary Fig. S5A). We then searched for CNV segment size variation in the global genome panel. We binned 1-kb CNV events into larger contiguous calls of presence or absence (Supplementary Data 5; Supplementary Fig. S5B). In the upper quartile (QA > 21), we found duplicated gene segments to be larger in size and encode more genes compared to deletions (Fig. 3E; Supplementary Fig. S5C). Taken together, our findings are consistent with CNVs being under purifying selection and that gene duplications show high population specificity.

We investigated features shared by CNV-affected genes and found that such genes were, on average, closer to TEs than conserved genes (Supplementary Fig. S6A). Furthermore, gene deletions were closer to TEs than duplications ($p$-value < 0.01; Fig. 3F). We then asked whether coding sequences of CNV genes carry more high-impact SNVs (i.e., variants with disruptive effects) compared to conserved genes. Genes segregating deletions harbored higher impact variants compared to conserved genes and genes with duplications. Common duplicated genes exhibit the highest number of high-impact variants (median 0.0012; Supplementary Fig. S6B), both consistent with functional redundancy (i.e., relaxed selection) and genomic defenses affecting coding sequences. We also found that deleted or partially deleted genes were enriched for H3K27me3 repressive histone methylation marks. In contrast, conserved genes and genes with duplications were enriched for H3K4me2 euchromatin marks (Supplementary Fig. S7A). Consistent with these observations, CNV genes show higher transcriptional variation during host infection and possibly higher functional redundancy (Supplementary Fig. S7B). Overall, CNV genes were functionally enriched in metabolic processes, including toxic secondary metabolic processes and peptidase activity (Fig. 3G; Supplementary Data 6; Supplementary Fig. S7C). Gene dispensability and functional enrichment of metabolic processes suggest that gene CNVs facilitate metabolic diversification and local adaptation.

**Effects of genome defenses and signatures of local adaptation**
Population differentiation across the globe detected at 218 gene CNVs is broadly congruent with SNV-based assessments of population differentiation[42] (Fig. 4A, Supplementary Fig. S8A). In Europe, the CNV-based population structure showed a pattern consistent with recent immigration events (Fig. 4A)[42]. In addition, the European population showed higher rates of gene flow with other regions (Supplementary Fig. S8B), corroborating the role the continent played in historic pathogen dispersal. Populations across the globe differed substantially in the rate of CNV events per strain (19−83 with a median of 44; Fig. 4B). Interestingly, we found a strong enrichment in duplications in the clusters assigned to Australia, NA−USA and Europe (Fig. 4B). An important factor shaping observed CNV rates among populations is the potential activity of RIP genomic defenses. Genomes with a functional copy of *dim2* likely express functional RIP machinery[42,50]. D*im2*-carrying populations tended to show higher rates of gene deletions (Fig. 4C). A mixed linear model accounting for population effects showed a weak but significant association of gene deletions and the strength of RIP ($r^2 = 0.023$; Supplementary Fig. S8C, D).

We identified candidates for local adaptation across all genetic clusters. We assessed the 95th upper quantile fixation index ($V_{ST}$) score per gene CNV among populations (Fig. 5A). The top $V_{ST}$ CNV gene (Zt09_13_00035) was predicted to be an effector gene with predicted functions in plant infection (Supplementary Data 7). The gene was rare and stable over time in the North African population but fixed across all other populations (Fig. 5B−D). Consistent with the predicted function, the gene is highly expressed during the initial stages of wheat infection and shares no homology outside of the species (Fig. 5E). Hence, the gene may play a role in adaptation to local host genotypes, favoring gene loss to avoid host recognition.

**Structural variation underpinning climate adaptation**
The species underwent climatic adaptation over the course of continental colonization[42]. We investigated the contributions of chromosome and gene CNVs to overall climatic adaptation using genome-environment association (GEA) analyses. We analyzed a total of 1099 samples and examined 19 bioclimatic factors (Supplementary Data 8; Supplementary Fig. S9A, B) based on two mixed-model association approaches for adaptive CNV discovery (Bonferroni α = 0.05). Several climatic factors showed strong positive correlations ($r > 0.8$, $p$-value < 0.000; Supplementary Fig. S9C). We identified significant associations for the chromosome CNVs of accessory chromosomes 15, 17, and 20 (Supplementary Fig. S10A; Supplementary Data 9). Chromosome 20 was the most consistently retrieved by both GEA methods, revealing the mean temperature to the coldest and driest quarters (Supplementary Fig. S10A). Next, we analyzed phenotypic trait variation for a subset of the global collection of strains ($n = 145$; Supplementary Data 10)[51], and no trait was significantly associated with CNVs (FDR 5% threshold). We identified 21 gene-level CNV associations with climatic factors spanning 14 different loci (Fig. 6A, Supplementary Data 11). Associated loci encoded functions, including epigenetic regulation, metabolism, and cell signaling functions (Supplementary Data 11). We found the strongest correlations with the climatic factors of the maximum temperature of the warmest month and the mean temperature of the warmest quarter (Fig. 6A; $r = 0.84$; $p$-value < 0.001, Supplementary Fig. S9C). The associated CNVs were segregating variable gene deletion frequencies (Fig. 6B).

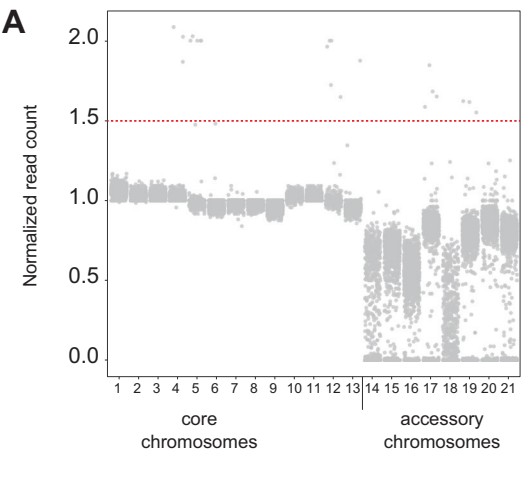

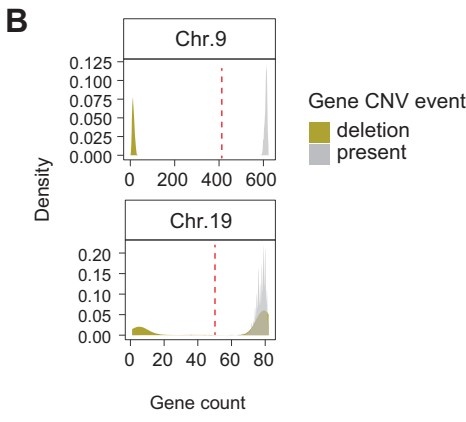

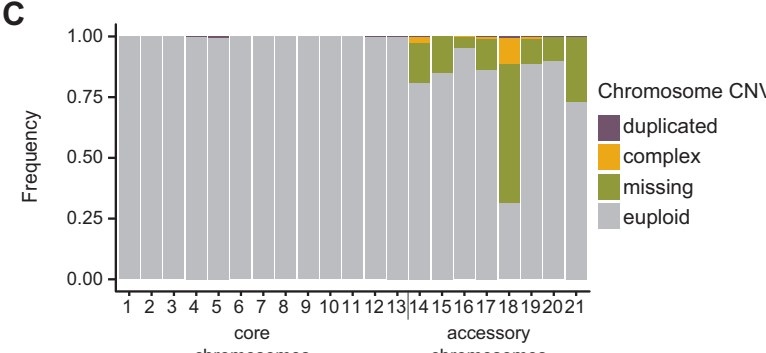

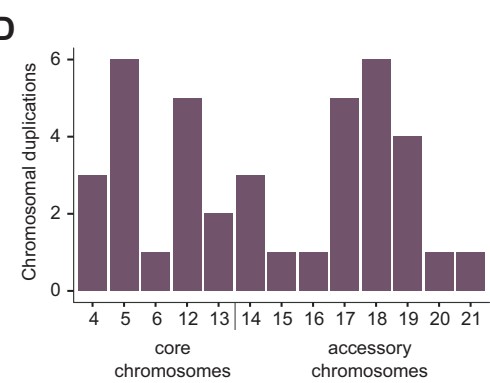

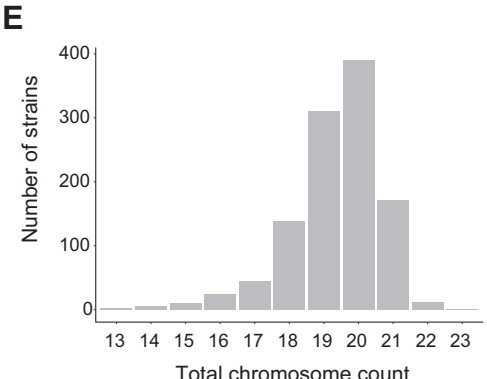

**Fig. 2 | Global survey of CNVs. A** Read coverage ratio between accessory chromosomes and core chromosomes in the global collection ($n = 1104$). Individual chromosomes exceeding 1.5 times the mean core chromosome coverage were categorized as being either fully or at least partially duplicated. Chromosomes with coverage near the threshold ($1.5 \pm 0.25$) were manually curated. **B** Distribution of gene presence/absence calls for the chromosomes 9 (core) and 19 (accessory). The 60% gene presence threshold in red was used to label the whole chromosome to be present or absent. **C** Frequency of chromosome CNVs in the global collection. Complex CNVs are defined as chromosomes exhibiting large deletions and duplications across chromosome arms. **D** Distribution of chromosome duplications (full and partial). **E** Total chromosome count including duplicates. Source data for Fig. 2A-E can be found in Supplementary Data 4.

We integrated CNV and SNV-based GEA analyses and identified three genes with shared evidence from both marker types (Supplementary Data 11)[42]. In general, gene functions identified by each variant type were only moderately overlapping ($r^2 < 0.6$, Supplementary Fig. S9C; Supplementary Data 11). This suggests largely independent contributions by CNVs and SNVs to climate adaptation. We identified a significant association in the gene CNV of Zt09_9_00561 encoding interferon 6 (Fig. 6A), which was supported by both association

mapping methods. Gene presence is associated with higher mean temperatures of the wettest quarter (Supplementary Data 11). We also found congruent mean annual temperature associations for the CNVs for the gene pair Zt09_2_00058/60 (Fig. 6A) and the SNV association at Zt09_2_00069 (Supplementary Data 11). The associated CNVs flank a biosynthetic gene cluster (BGC19) on chromosome 2 with the second highest fixation index values ($V_{ST}$) of all gene CNVs. (Figs. 5A, 6C; Supplementary Fig. S10B). We investigated the nature of BGC19 and found

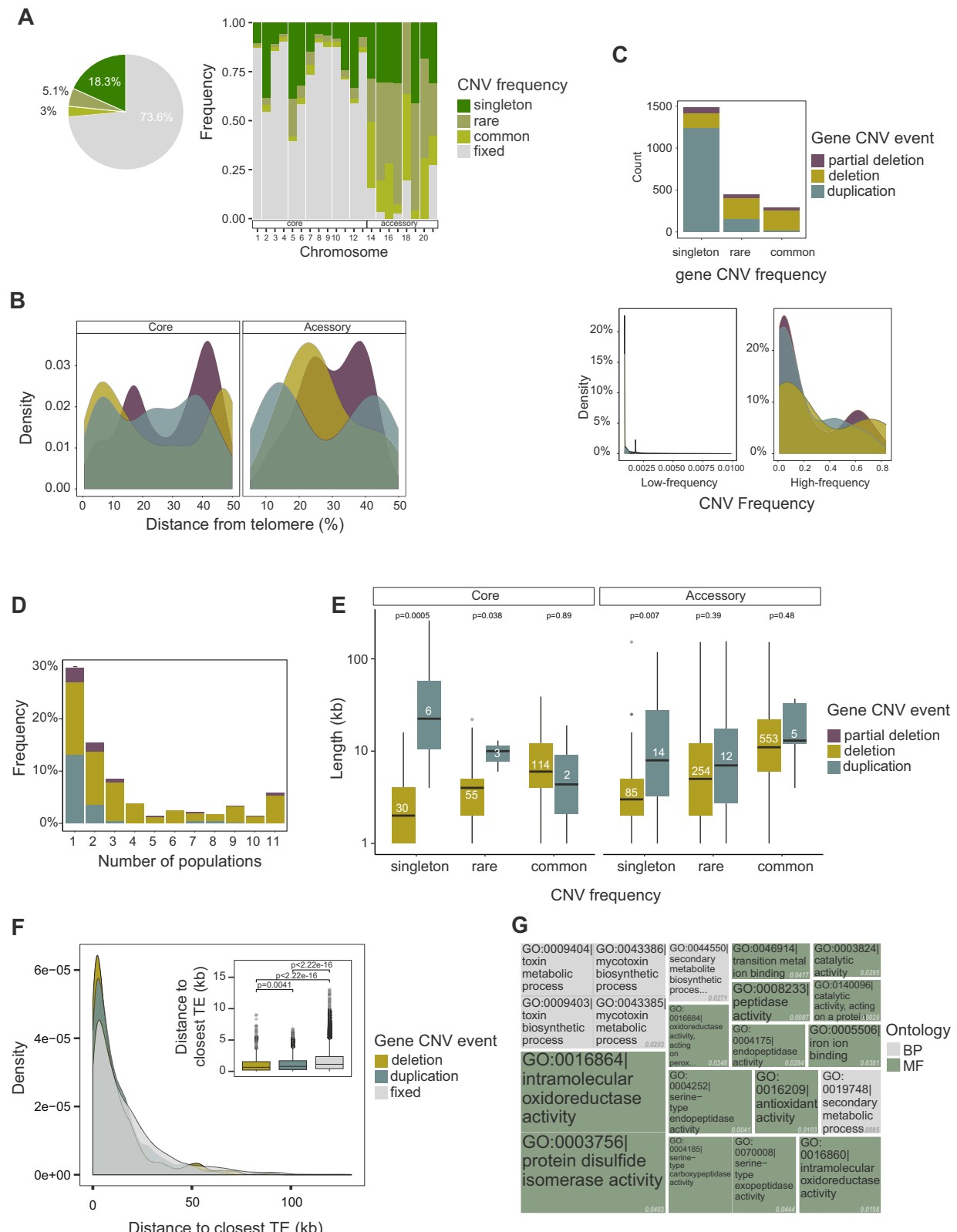

that it is a 63-kb cluster of unknown function (Fig. 6C). The BGC is present in the sister species *Z. brevis*, and the loss of the core biosynthetic gene (Zto9_2_00067) was confirmed by PCR[47]. The presence of BGC19 was negatively correlated with mean annual temperatures ($r = -0.61$, $p$-value < 2.2e−16; Fig. 6D and E). The loss of the BGC was also associated with higher reproduction on wheat leaves of 12 wheat cultivars (Fig. 6F; Supplementary Data 10). Taken together, the deletion of

the BGC19 gene cluster is likely under antagonistic pleiotropy for high-temperature adaptation and host colonization potential.

## The climate-associated Sir2 locus is occupied by a massive Starship mobile element

Causal factors contributing to environmental adaptation remain poorly understood. We took advantage of the high-quality pangenome

**Fig. 3 | Population genetic features of gene CNVs. A** Pie chart of the overall percentage of gene CNV frequency in the global collection. The bar plot shows the gene CNV frequency per chromosome. Genes without evidence for any CNV event are grouped in the category CNV frequency fixed. The analysis was performed solely with euploid chromosomes to remove any bias from missing or duplicated chromosomes. **B** Density plot showing an overview of CNV frequency types across chromosome arms. Colors refer to gene CNV events. **C** Gene CNV event of each CNV frequency type. The density plot shows the CNV event distribution of low-frequency (singleton and rare frequency type) versus high-frequency (frequencies > 1%) CNV. **D** Number of gene CNVs shared between the 11 genetic clusters (i.e. populations). Colors refer to gene CNV events. Source data can be found in Supplementary Data 4. **E** CNV segment size across different CNV events and

frequencies in the highest quartile CNV segment quality score (QA). The number within the boxplot refers to the sample size. *N* = 1137. Pairwise two-sided Wilcoxon test. Values on top refer to adjusted *p* value (Holm method). Groups with <3 samples lacked statistical power. Source data can be found in Supplementary Data 5. **F** CNV gene distance to the closest transposable element. The box center line represents the median, and the limits represent the first and third quartiles. Whiskers indicate maximum and minimum values. *N* = 8164. Pairwise two-side Wilcoxon test, adjusted *p*-value (Holm method). **G** Gene ontology enrichment analysis of CNV genes. Values in gray refer to two-sided Fisher's test *p* values. BP refers to biological process and MF refers to molecular function. Source data can be found in Supplementary Data 6.

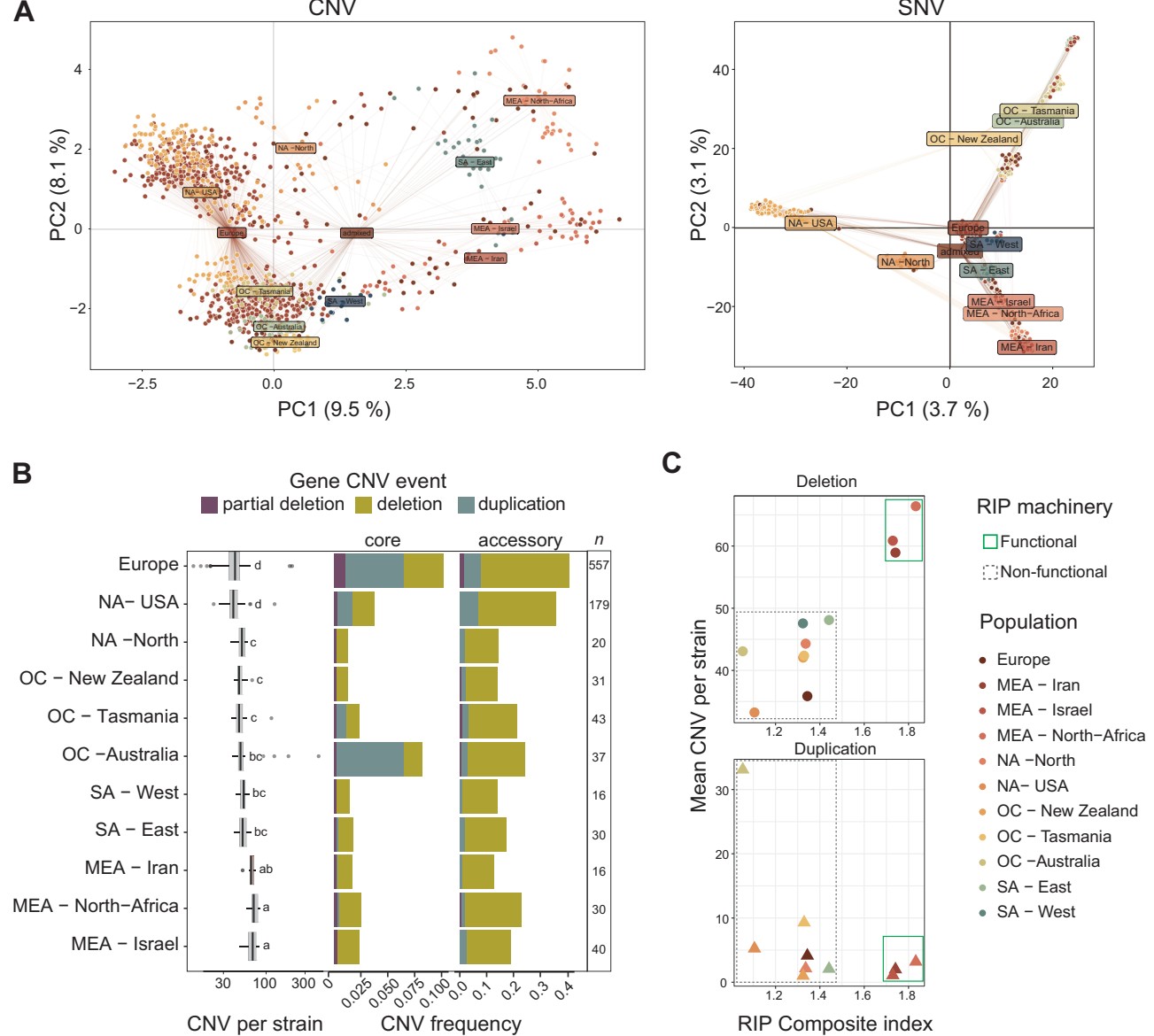

**Fig. 4 | Global copy-number polymorphism population structure. A** Principal component (PC) analyses based on 218 CNV genes (left panel) and 26,728 single nucleotide variants (SNVs; right panel). The color scheme identifies the genetic clusters (see Fig. 1). **B** Number of gene CNVs per strain and variant frequencies at CNV loci per population in core and accessory chromosomes. The sample number per population is shown in the last column. Different letters above the boxplots per population indicate significantly different groups based on a one-way ANOVA followed by Tukey's test.

identify significantly (adjusted *p*-value < 0.05) different groups based on a one-way ANOVA followed by Tukey's test. *N* = 1104 samples. **C** Relationship between mean gene CNV number per strain and RIP composite index[42] in gene deletions and duplications. Genomes were labeled as carrying a functional RIP machinery if an intact copy of the *dim2* methyltransferase gene underpinning activation of RIP was detected. Source data for **A** and **B** can be found in Supplementary Data 4.

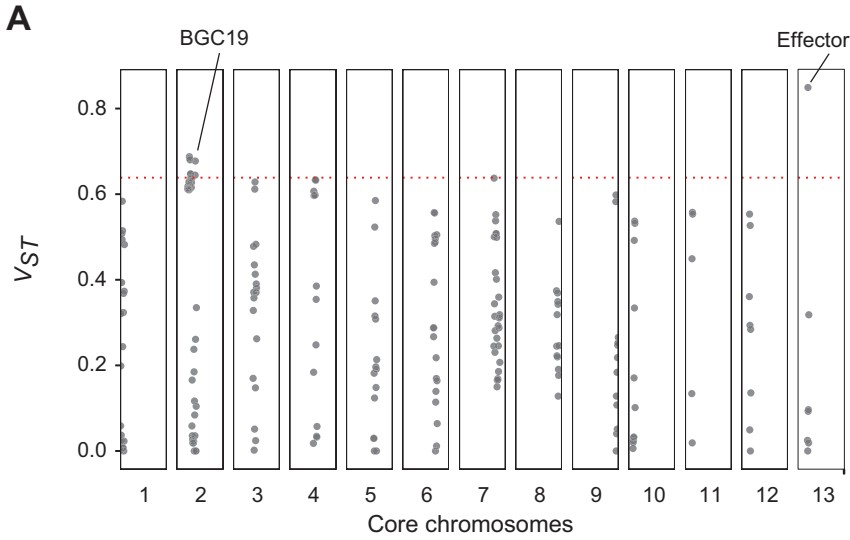

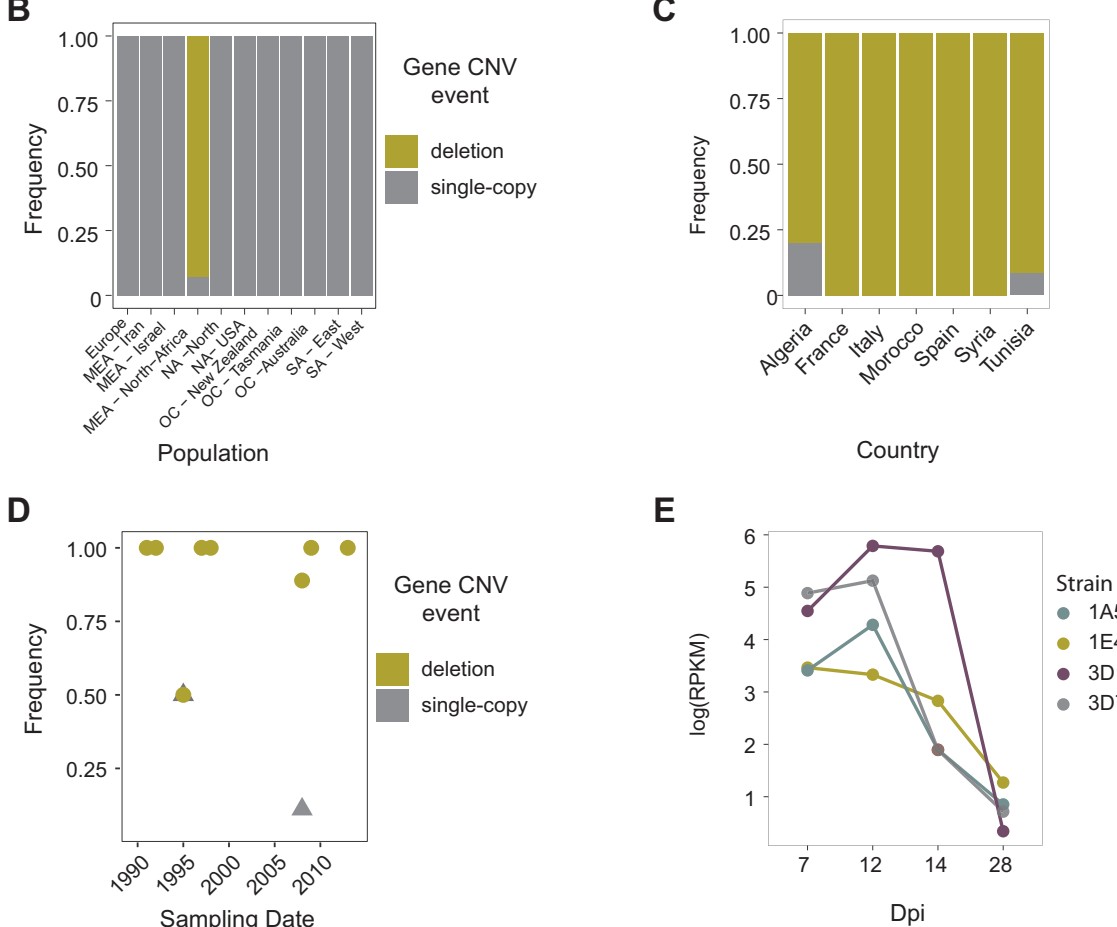

**Fig. 5 | Signatures of local adaptation. A** CNV variant differentiation based on the fixation index $V_{ST}$ across core chromosomes. $N = 218$ genes. The red dotted line marks the 95th quantile used as a threshold. Source data can be found in Supplementary Data 7. **B** CNV frequencies of the top $V_{ST}$ CNV locus across populations. **C** Frequency variation at the effector locus in MEA-North Africa genomes and **D** changes over time in MEA genomes. **E** Expression profile[155] of the effector gene during host infection in four strains (dpi: days post-infection). Source data for panels **C** and **D** can be found in Supplementary Data 4.

resources for the species to investigate the significant association of the annual temperature range with a CNV on chromosome 7 (Figs. 6A, 7A). The gene Zt09_7_00034 is predicted to encode a homolog of *Sir2*, an NAD-dependent deacetylase major protein family associated with lifespan and mating type silencing in yeast, as well as aging in humans[52–54]. A phylogenetic analysis of the Sirtuin family showed that *Z. tritici Sir2* is an ancient duplication of *Sir5* in Dothideomycetes followed by multiple independent gene losses (Supplementary Fig. S11).

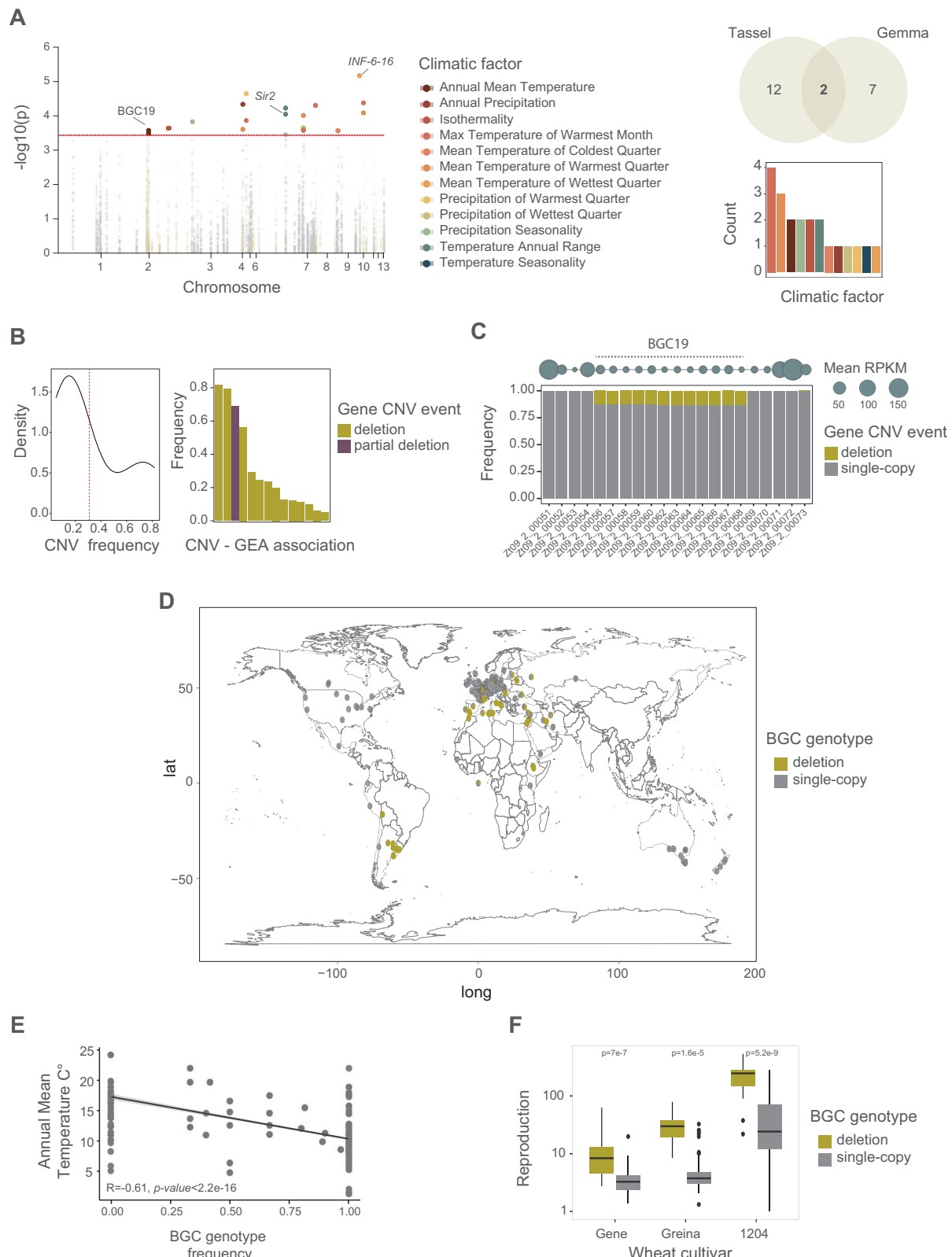

To investigate the chromosomal context of the *Sir2* deletion, we analyzed both unfiltered CNV call data and chromosome-level assembled genomes (*n* = 19). CNV frequency analyses of the global collection showed that the region segregates a large insertion encoding transcriptionally active genes as well as TEs (Fig. 7A). The *Sir2* locus was partially present in the center of origin populations (i.e., Middle East, Iran) and fixed in Oceania and the USA (Supplementary Fig. S12A). The

gene Zt09_7_00033 adjacent to *Sir2* encodes a DUF3435 domain characteristic of a newly described family of tyrosine recombinases[55]. This fungal-specific tyrosine domain is required for the mobilization of massive TEs identified as *Starships*[55,56]. Genes unrelated to DUF3435 encoded inside the *Starship* are called cargo genes. The massive mobile element resides close to the major global regulator Velvet[57], which is linked to sexual reproduction and growth[58]. We found a single

**Fig. 6 | CNV contribution to environmental range adaptation. A** Manhattan plot of the genome-environment association (GEA) analyses using two mixed model methods (Tassel and GEMMA). The red dotted line refers to the Bonferroni (alpha = 0.05) threshold. The Venn diagram shows the GEA hit overlap between mixed model methods (Tassel and GEMMA). The bar plot shows the number of significant hits per climatic factor. Colors refer to climatic factors. Source data can be found in Supplementary S11. **B** Distribution of gene CNV frequency and significant GEA hits in the global collection. The red dotted line refers to the mean frequency. Source data can be found in Supplementary Data 8. **C** Biosynthetic gene cluster (BGC) 19 frequency in the global collection. Circles at the top refer to mean transcriptional activity under in vitro conditions in a subset of the global collection

($n = 74$). **D** Presence of BGC19 across geography. $N = 1104$. The world map was created with the R package *maps* v. 3.4.2 based on Natural Earth project data (https://www.naturalearthdata.com). **E** Correlation of BGC presence with the associated phenotype (annual mean temperature). Two-sided Pearson correlation test. Source data can be found in Supplementary Data 4. **F** On-host reproduction rate in a subset ($n = 145$) of global collection grouped by presence/absence of BGC19 and three different wheat cultivars. Data displayed the overlapping hits using Tassel and GEMMA. The box center line represents the median, and the limits represent the first and third quartiles. Whiskers indicate maximum and minimum values. $N = 130$ samples per cultivar. Pairwise two-sided Wilcoxon test, adjusted $p$ value (Holm method). Source data can be found in Supplementary Data 10.

insertion in the *Starship* (hereafter identified as *Swordfish*) present mostly as a single copy (global frequency of 64.5%) or entirely missing (14.3% globally), and additional haplotypes showed cargo gene variation (Supplementary Fig. S12B). *Swordfish* boundaries contained no detectable direct repeats flanking the element, suggesting that the element lost the ability to transpose. The *Swordfish* region is rich in TEs (Fig. 7A), and chromosome-level assembled genomes revealed syntenic *Swordfish* flanks among sister species (Supplementary Fig. S13A). Strains harboring *Starship* carry ~68k additional sequences (SD = 13.5 kb; Supplementary Fig. S13B).

We used gene synteny and phylogenetic analysis to retrace the evolution of *Swordfish*. Phylogenies of the flanking genes match the evolutionary history of the genus (Supplementary Fig. S14). *Swordfish* gene cargo underwent a complex sequence of duplications, transposition, and multiple, independent gene losses (Fig. 7B, Supplementary Fig. S15A). For example, the genes Zt09_7_00039/38 have paralogs in different genetic contexts, suggesting ancestral duplications (Supplementary Fig. S15A). Chromosome-level assemblies lacking *Swordfish* (strains IR01_48b, CNR93, UR95) show the flanking gene Zt09_7_00030 and cargo genes Zt09_7_00037/38 in a different location on chromosome 7. Although the *Swordfish* is never present in more than one copy, a close homolog of the *Starship* tyrosine recombinase was found in strains lacking *Starship* (i.e., strain Zt269). Distant duplicates of the tyrosine recombinase (identity < 60%) were found on chromosomes 1 and 12 (Supplementary Fig. S15B). Strain 3D1 carried a tandem duplication of the cargo genes Zt09_7_00033/34/35 (Supplementary Fig. S13A). *Swordfish* is rich in TE sequences, including specific retrotransposons (RLX_LARD_Gridr and RII_Philae; Supplementary Fig. S16A, B). Some strains lacking *Swordfish* carry cargo genes (Zt09_7_00037-39) in a different genomic location, suggesting that *Swordfish* cargo turnover was recent and possibly mediated by TEs (Supplementary Fig. S16C). Epigenetic analyses revealed a bi-repressive pattern regulated by H3K27me3 and H3k9me2 modifications (Fig. 7C). Flanking regions are marked by H3K4me2 euchromatin. The predicted secreted protein Zt09_07_00037 and the flanking gene Zt09_7_00040 are highly expressed during infection (Fig. 7C). We found a high degree of sequence conservation among the triplet Zt09_7_00034/35 lacking coding sequence variants, suggesting strong purifying selection and conservation of synteny (Fig. 7C, Supplementary Fig. S17).

The *Swordfish* cargo gene *Sir2* CNV is significantly associated with the annual range of temperature, and the neighboring highly expressed secreted gene Zt09_7_00037 is associated with the mean diurnal range. Both climatic factors are moderately correlated ($r = 0.55$, $p$-value < 0.001). Overall, we identified a possible new role for *Sir2* as a factor in climate adaptation. Hence, the massive *swordfish* mobile element likely contributes to thermal adaptation and shapes the species range across highly variable environmental conditions.

## Discussion

Climatic factors are strong determinants of pathogen spread and disease severity[39,59]. Genetic variability provides the substrate for rapid adaptation to a changing environment[60] and to cope with

environmental stressors[61–63]. How copy number variation shapes climate gradients spanned by individual species remains poorly understood. We show that multiple gene deletions contribute to pathogen adaptation across continental climatic gradients. CNVs are also drivers of metabolic diversity, including contributions by *Starship* mobile elements reshuffling genes carried as cargo.

Approximately a quarter of all genes in the pathogen species were affected by CNV events, likely reflecting an equilibrium between new CNVs being generated and purifying selection acting against CNVs[64]. Genes segregating CNVs were located closer to TEs and were functionally enriched for catalytic activities and secondary metabolic processes compared to more conserved genes. Furthermore, gene duplications were the most abundant CNV events yet remained at low frequency in populations and were rarely shared among populations. The skew in CNV events may stem from a duplication detection bias; however, a more stringent control for call quality and CNV frequencies produced similar outcomes. Gene duplications are a powerful source for gene neofunctionalization[65–67] and promote non-allelic homologous recombination due to homology among duplicates[68,69]. *Z. tritici* is a highly recombinant species[70,71], suggesting that segmental duplications could impact the likelihood of nonallelic homologous recombination[69]. The evolutionary history of the pathogen has likely impacted the rates of duplications on chromosomes, as observed in the more recently colonized Oceanian and American continents. More recent populations exhibited signatures of bottlenecks with reduced genetic diversity[42]. Concurrently, the bottlenecked populations experienced an increase in TE activity, most likely caused by a loss of defense mechanisms against TEs[42,50]. RIP triggers rapid mutation accumulation after duplication, leading to loss-of-function or, more rarely, diversifying selection in populations[72,73]. RIP activity likely acted as a driver of gene loss by rapidly mutating genes and facilitating the purging of nonfunctional copies through excision. Such nonrandom processes underlying the creation and elimination of structural variation illuminate how species gene pools can evolve over short evolutionary time periods.

The role of gene duplications in environmental adaptation is well documented across kingdoms[66,74,75]. How gene loss can contribute to adaptation is less well understood. Our analyses are supporting previous work showing that gene deletions are under strong purifying selection in *Z. tritici* populations[47]. Furthermore, gene losses can largely be explained CNV-driven environmental adaptation. We found signatures for adaptive gene loss ranging from single genes to chromosome copy number variation. Gene loss typically arises from the abrupt rearrangement of coding sequences by repetitive elements, unequal crossing-over events, or by the accumulation of mutations leading to a loss of function[76,77]. For example, loss of the *Desat2* gene in cosmopolitan *D. melanogaster* was linked to resistance to cold[78]. Hence, adaptive loss to changing environments shows convergence across kingdoms[77,79–82].

We found strong associations indicating that climate adaptation across continents was likely facilitated by the presence/absence of variation in a biosynthetic gene cluster. The presence of the BGC19 cluster is positively associated with colder climates, and cluster

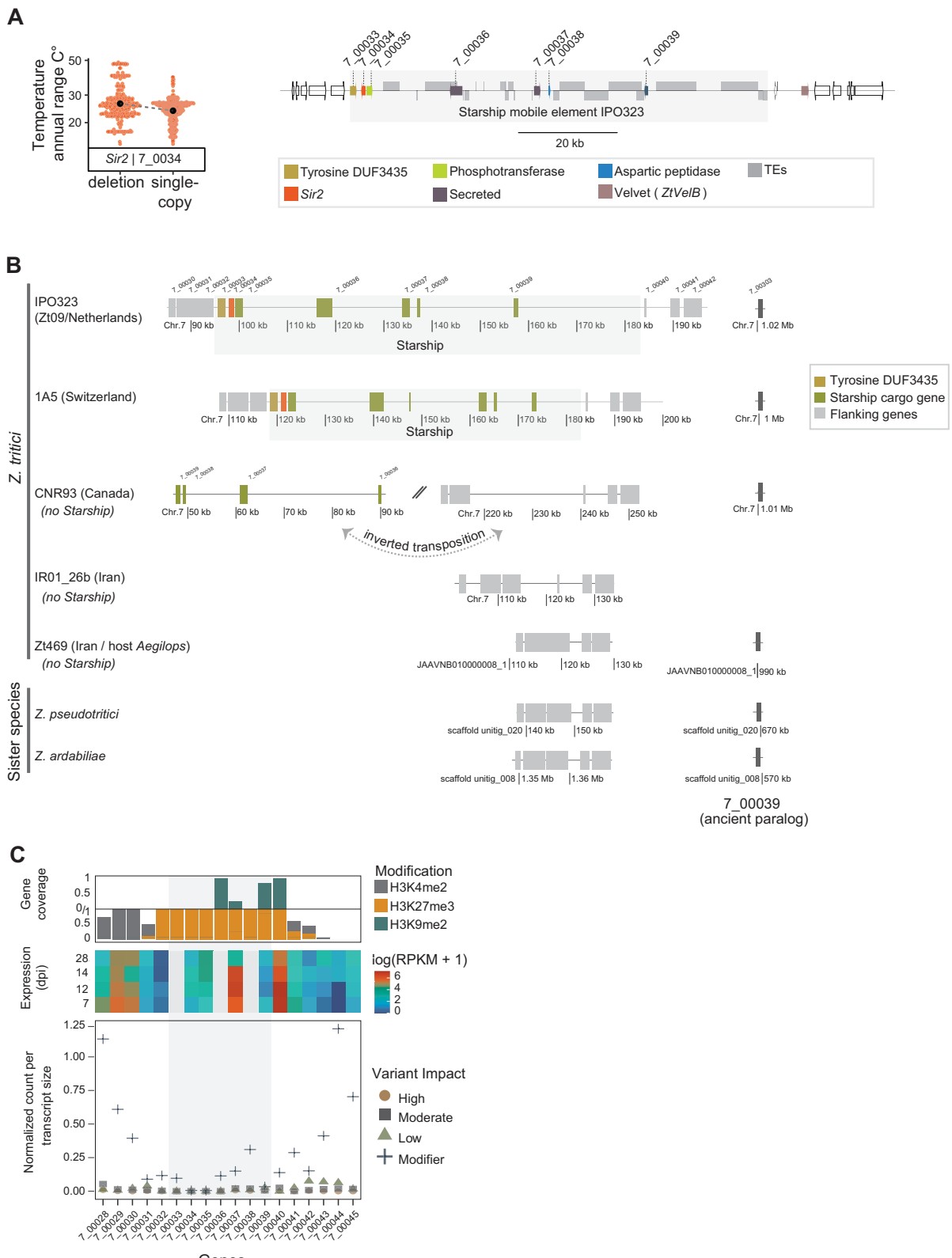

**Fig. 7 | *Starship* mobile element associated with thermal adaptation.**
**A** Temperature variation associated with the CNV of *Sir2*. *Starship* gene order in the reference genome IPO323. The black circle represents the median. *N* = 920 samples. Source data can be found in Supplementary Data 8. **B** Synteny plot between chromosome-level assemblies of *Z. tritici* strains and sister species within the *Starship* region. Diagonal bars refer to different genomic contexts, and dotted arrows refer to transposition events between regions. **C** Predicted impact on protein sequences (normalized by transcript size), transcriptional activity during the infection host cycle, and post-translational modifications for *Starship* and flanking region genes. Dpi: days post-infection. Expression data[126] includes three biological replicates per condition (i.e. dpi). The light gray refers to the *Starship* cargo genes.

deletion is associated with an increased reproduction rate in specific wheat cultivars. Fungal BGCs are a major source of chemical diversity, and tight physical organization improves coregulation efficiency and reduces cytotoxicity caused by intermediary products[83]. Secondary metabolites are involved in numerous cellular functions, including virulence, defense, and growth[84]. BGCs can be hotspots generating structural variation[85–87] and favor adaptation[88,89]. Adaptive loss of BGCs is thought to be explained by the Black Queen hypothesis, which argues that communities sharing "leaky" resources such as metabolites will favor genome reduction and loss of gene sets necessary to make such metabolites accessible[90]. As we have identified linked antagonistic pleiotropy governing the presence of BGC19, the gene cluster may be under a complex selection regime across populations.

In addition to climate adaptation, virulence factors (i.e., effectors) can also undergo adaptive gene loss to facilitate pathogen adaptation[52,91]. In an arms race between host and pathogens, such virulence factors can be important triggers for host defense mechanisms or serve to manipulate the immune response[91,92]. The sampling in North Africa covered a 30-year timespan and showed consistently low frequencies of the virulence factor in populations. The region predominantly produces durum wheat (*Triticum durum*) in contrast to the more widely cultivated bread wheat (*T. aestivum*)[93]. Strong selection in North African populations was likely driven by pathogen-specific recognition mechanisms[94,95]. Overall, we found that gene loss at different scales likely played an important role in environmental adaptation across the historical spread of the pathogen population.

We identified a *Starship* mobile element associated with climate adaptation in fungal populations. *Starships* are widespread among fungi and were recently characterized as tyrosine-recombinase-mobilized DNA transposons amassing multiple host genes and TEs[56,96]. Elements can reach up to approximately half a megabase in size and carry species-specific cargo genes. Individual *Starships* were associated with adaptive functions such as heavy metal tolerance in a strain-specific manner[97] spore killing-mediated meiotic drive[96], and formaldehyde resistance[98]. *Starships* are powerful agents that reshuffle the core functions of gene clusters[98]. The cargo carried by *swordfish* expands our understanding of adaptive functions associated with massive selfish elements. The cargo includes gene CNVs associated with thermal range climatic factors that may contribute to the environmental range of pathogen populations[19,99,100]. Signatures of thermal adaptation were previously documented in *Z. tritici* without identifying a molecular mechanism[101–103]. Surprisingly, the *Swordfish* element resides within a highly conserved region across sister species and neighbors the gene *VeltB*, which is a key regulator of reproduction, metabolism, and growth processes in fungi[57]. The strong epigenetic repression across the entire *Swordfish* element is consistent with the strong repression of TE activity immediately adjacent to a transcriptionally open and conserved region of the chromosome[104]. We hypothesize that this *Starship* likely inserted itself at this locus after the emergence of the species, followed by a complex set of duplications and transposition events facilitated by high TE activity in the region.

*Swordfish* carries an ancient duplication of the *Sir5* gene. The sirtuin protein family is a group of evolutionarily conserved NAD+-dependent histone deacetylases involved in regulating epigenetic processes and[105] and through their enzymatic activity, sirtuins modulate the acetylation status of histones and other proteins, which have been implicated in a diverse range of biological processes in eukaryotes, including cellular aging, diseases, genome stability, metabolism, and stress response[52]. The Zt09_7_00034 sirtuin resides exclusively next to the tyrosine recombinase gene within the *Swordfish*. The lack of SNV variants in the coding sequence, coupled with transcription activity, suggests a local regulatory role within the epigenetic landscape of the element, partially explaining the successful insertion of the element in a highly conserved genomic region. In fungi, sirtuins are involved in mating-type loci silencing, rDNA stabilization, host defense suppression, and secondary metabolism[54,106,107]. Here, we found evidence of a sirtuin protein linked to climate adaptation in fungi.

We retraced the *Starship* element formation and hypothesize that *Swordfish* was formed after speciation and spread across the globe following duplication events, diversification, and retrotransposon-mediated gene transposition. Some pathogen populations nearly fixed or lost *Swordfish*. The absence of direct repeats at the flanks of the element is consistent with full integration into the fungal genome. The integration followed by mobility impairment suggests that Swordfish was domesticated after its insertion at this locus through selection for adaptative gene cargo and selection against deleterious effects of mobile element activity. Overall, we show that CNVs can be drivers of metabolic diversity and contribute to the global climate adaptation of a crop pathogen. We found that rates of gene loss were strongly associated with the efficiency of genomic defenses, influence adaptive loss-of-functions and prevent maladaptation to a changing environment[108]. Large genome panels enable the retracing of pathogens spread across continents and disentangle random effects from the most likely effects of local adaptation, highlighting the role CNVs play in the evolution of microbial species.

## Methods

### Sampling

We performed copy number variation analysis on a global collection of *Zymoseptoria tritici* comprising 1109 Illumina short-read genomes (Fig. 1A). The collection covers strains originating from 42 countries representative of the history of wheat domestication and historical expansion of wheat cultivation (Supplementary Data 1)[42]. Additionally, we used high-quality full chromosome assemblies based on PacBio long reads of 19 strains of *Z. tritici* representative of the global genetic diversity of the species and genomes of four sister species (*Z. pseudotritici, Z. ardabilae, Z. brevis* and *Z. passerinii*)[44,109] for CNV call validation. Information about sample origin, sequence quality, and accession numbers is provided in Supplementary Data 1.

### CNV calling

Illumina raw reads were trimmed with Trimmomatic v. 032[110] and mapped to the *Z. tritici* (IPO323) reference genome using Bowtie2 v.2.4.0[111] very-sensitive-local parameter. We used GATK CNV caller v.4.1.9.0[112] with recommended parameters on alignment BAM files ($n$ = 1109). The software scans read coverage and models sequencing biases based on negative binomials, taking copy number states and genomic regions of CNV activity into account for a hierarchical hidden Markov model (HHMM). We set the CNV interval to 1000 bp windows with no overlap. Such intervals are recommended to account for variation in sequence coverage (Supplementary Data 1). We filtered for GC content in windows (min = 0.1 and max = 0.9), as well as extremely low and high read counts (--low-count-filter-count-threshold = 5, --extreme-count-filter-minimum-percentile = 1, --extreme-count-filter-maximum-percentile = 99). We then built a prior table for chromosomal ploidy to assign prior probabilities for each ploidy state. Finally, we called CNV genotypes using the germlineCNVCaller and PostprocessGermlineCNVCalls functions. After genome-wide CNV calling, we filtered and validated gene CNVs. We used bedtools v2.31[113] annotate to overlap gene elements with the CNV calling.

### CNV filtering and validation

**CNV coherence validation.** To assess CNV calling quality, we analyzed seven pairs of independently sequenced strain (i.e., the same strain with at least two independent library preparation and sequencing efforts; Supplementary Data 2). After validation, we retained the strain with the higher mean coverage of each replicate pair in the dataset for further analysis.

**Gene CNV events**. CNV calling was performed in 1-kb windows across the genome as described above (see "CNV calling"), allowing for ambiguous calls in polymorphic gene elements. Deletion, duplication and single-copy CNV events were attributed to a gene if the event coverage was >80% of the gene. We defined partial deletions if the deletion covered 50–80% of the gene. Additional combinations were defined as single-copy events.

**CNV filtering and validation.** To reduce false positive calls in the dataset, we used quality scores (CNQ), which are defined as the difference between the two best genotypes, Phred-scaled log posteriors. We set the CNQ threshold based on the structural variant calling verified using the fully assembled genomes of *Z. tritici*[44]. We contrasted CNV calls by the GATK calling pipeline to pairwise whole-genome comparisons of chromosome-level assemblies based on the software SyRI v1.3[114] using IPO323 as the reference genome. For a direct comparison of variant genotyping, we analyzed four strains present in both the global dataset (Illumina reads) and the chromosome-level assembly dataset (PacBio reads; samples 3D1, 3D7, 1E4, and 1A5). We first subset gene sets to unambiguous CNV calls (i.e., with more than 60% of event coverage) in both datasets. We then subset the SyRI structural variant calls to single-copy regions, translocation, and DNA gain and loss, removing SNV calls (Supplementary Data 3). We compared the call quality based on CNQ to the gCNV GATK output. The levels of matching and discordant calls between tools were used to define thresholds for CNV GATK calls. We filtered for missing data in the global dataset by removing calls with <50% coverage or <80% call frequency. To define CNV segments (i.e., larger regions with consistent CN calls), we binned GATK CNV Caller segment output per strain and calculated the CNV segment quality call QA defined as complementary Phred-scaled probability at all points (i.e., bins) in the segment, which agree with the segment copy-number call. We then subset for CNV segments based on the filtered dataset and kept CNV segments within the upper quartile quality. We retrieved 14 loci to cross-check the validation with a PCR screen performed earlier for deletion polymorphisms[47].

**Chromosome number variation.** *Z. tritici* is haploid and carries 13 core chromosomes and up to eight accessory chromosomes (not shared by all members of the species). We used chromosome-wide copy number estimates using the median coverage (MAPQ > 20) of core chromosomes for each strain to define accessory chromosome presence/absence among all strains. Core chromosomes with more than 1.5 times the core coverage were defined as likely duplicated. Chromosome coverage close to the cutoff threshold ($1.5 \pm 0.3$) was manually curated. The high polymorphism of accessory chromosomes[44] makes the implementation of thresholds challenging. We defined accessory chromosome absence with gene presence falling below 60% and chromosome duplication with >60% gene duplications. To validate the utility of these thresholds, we analyzed eight strains included in this study for which chromosome-level assemblies are available (3D1, 3D7, 1E4, 1A5, 08STF040, 08STCZ015, 08STCH015 and 09STD078)[44].

**Single nucleotide variant call and de novo genome assembly**
To analyze the genomic context of CNVs in the population, we retrieved SNV calls and de novo genome assembly analyses previously performed for the same global collection ($n = 1109$)[42]. Briefly, reads were mapped to the reference genome (IPO323) using Bowtie2 v.2.4.139[111]. SNVs and short indels were assessed using the short-variant pipeline performed with GATK v4.1.4. HaplotypeCaller[115]. Ploidy was set to 1, and hard filtering was performed. The per-site filters included FS > 10, 444 MQ < 20, QD < 20, ReadPosRankSum between −2 and 2, MQRankSum between −2 and 2, and BaseQRankSum between −2 and 2. We filtered the dataset for biallelic SNV genotypes, a minor allele frequency of 0.05, and max missing genotype data of 20%. We further

used the dataset to predict the effect of SNVs on encoded proteins (categories high, moderate, low, and modifier) using SnpEff version 4.3[116]. We filtered for CNV genes with a ≥80% single copy genotype in the global panel. We built a database based on the reference genome IPO323 annotation and filtered for the top impact effect per variant. De novo assemblies were generated using SPADES v3.14.1 software[117] with the "--careful" parameter to reduce mismatches.

**Population structure analysis**
The global collection included 1003 genomes grouped into 11 genetic clusters and 106 admixed samples defined as showing <75% assignment to any specific cluster[42]. CNV-based population structure and *VST* outlier analyses were performed on the filtered, biallelic gene CNV dataset with a single copy defined as the reference allele and the most frequently observed CNV event at the locus as the alternative allele. We filtered core chromosome gene CNVs for a minor allele frequency ≥1% and 20% maximum missing genotypes per locus. The SNV-based PCA was generated using the SNV data subset with the BCFtools[118] "M2" option to keep only biallelic SNV and "-q 0.05:minor" for a minor allele frequency filter of 5%. We used vcftools[119] "--thin 1000" to keep only 1 SNV for every 1 kb interval. The PCA was performed with the ade4 R package v1.7-22[120] and visualized with the ggplot2 v3.4.2 R package[121]. CNV fixation indices ($V_{ST}$) were calculated using the *hierfstat* v0.5-11 R package[122].

**Environmental and life-history trait adaptation analysis**
We performed whole-chromosome CNV and gene CNV genotype-environment association (GEA) analyses using bioclimatic factors from the WorldClim database v.2.1 at 10' resolution. The bioclimatic data comprise historic averages (between 1970-2000) of 19 climatic variables related to temperature and precipitation. Geographic coordinates of strain collection sites were used to approximate climatic dataset gridpoints. We used two widely used mixed linear model association mapping tools: GEMMA v.0.98.3[123] and Tassel v5[124] using the Rtassel R package v0.9.29[125]. We used the thinned SNV dataset (see the section "Population structure") to estimate the kinship matrix to account for non-random relatedness among genotypes and reduce spurious associations. We contrasted our CNV-based GEA with the SNP-based GEA performed by Feurty et al.[42] using identical climatic datasets and tools. We used Bonferroni corrections (alpha = 0.05) to account for multiple testing in GEA analyses. We performed phenotype-genotype GWAS analyses on a subset of the global collection ($n = 145$ strains) using 24 life-history phenotypes[51]. The phenotypic data comprised virulence (i.e., lesion size) and reproduction (i.e., pycnidia production) of individual strains during host infection on 12 different wheat cultivars. Tests were conducted with GEMMA and Tassel software using the same parameters as for the GEA analyses.

**Transcriptome profiling**
We analyzed transcriptional profiles based on gene expression in minimal medium conditions of 19 *Z. tritici* chromosome-level assembly strains[44] and a collection of strains from a Swiss field population (Eschikon, Switzerland; $n = 74$)[48]. In summary, 10e5 cells were inoculated using liquid minimum media (MM) with a limited carbon source and grown for 7-10 days to reach the hyphal growth stage. RNA extraction was performed using a NucleoSpin RNA Plant Kit following the manufacturer's instructions[44]. We analyzed the publicly available transcriptome dataset (NCBI SRA accession SRP077418) of four strains also included in this study (3D1, 3D7, 1E4, and 1A5) inoculated on wheat plants and analyzed 7, 12, 14, and 28 days after infection[126]. Illumina raw sequencing reads were trimmed and filtered for adapter contamination using Trimmomatic v.0.32 (parameters: ILLUMINACLIP:-Trueseq3_PE.fa:2:30:10 LEADING:3 TRAILING:3 SLIDINGWINDOW:4:15 MINLEN:36)[110]. Filtered reads were aligned using HISAT2 v. 2.0.4 with default parameters[127] to the *Z. tritici* reference genome (IPO323). Mapped transcripts were quantified using HTSeq-count v.2.0.2[128]. Read

counts were normalized by calculating trimmed means of *M*-values (TMM) with the calcNormFactors option. To account for gene length, we calculated reads per kilobase per million mapped reads (RPKM) values using the R package *edgeR* v.3.42.2[129].

## Orthology analyses and characterization of gene functions

Orthologs were searched among the chromosome-level assemblies ($n = 19$) and four sister species of *Z. tritici* with Orthofinder v.2.2[130]. Orthologs shared between *Z. tritici* and at least one sister species were defined as conserved and used to infer gene losses versus gains in *Z. tritici*. TE annotations of the chromosome-level assemblies were retrieved from refs. 44,45. Chromosome-level assemblies of *Z. tritici* were also analyzed to predict biosynthesis-related gene clusters (BGCs) using antiSMASH v.5.0[131]. Identified gene clusters were further annotated using InterProScan v.5.54[132]. GO term enrichment analyses were performed using Fisher's exact tests based on gene counts with the *topGO* R package[133]. The GO term treemap was plotted using the *treemapify* R package[134]. We retrieved publicly available ChIP-seq datasets of histone modifications H3K4me2, H3K9m2, and H3K27me3 from the NCBI SRA (SRP059394) of the *Z. tritici* IPO323 reference genome isolate grown in rich medium[135]. ChIP-seq reads were trimmed with Trimmomatic v.0.32[110] and mapped to the IPO323 reference genome with Bowtie2 v.2.4[111]. Alignment BAM files were converted using BEDtools v.2.30[113,] and peak calling was performed using Homer v.4.11[136]. Gene coverage was analyzed with the BEDtools intersect command.

## Starship characterization and annotation

To characterize *Starship* mobile elements in the *Z. tritici* global collection, we searched for genes with CNVs for *Starship*-associated functional domains[56] using the hmmsearch function of HMMER v3.3.2 (*E*-value ≤ 0.001)[137]. We focused on genes belonging to a newly described family of tyrosine recombinases with DUF3435 domains (Protein Family accession: PF11917) that are both necessary and sufficient for the movement of entire elements[98]. We defined the boundaries of candidate elements by annotating their putative empty insertion sites. We aligned 25 kb upstream of the candidate tyrosine recombinases to the corresponding homologous region in isolates that were missing the gene to determine the upstream element boundary and then aligned 25 kb downstream of the homologous region back to the contig containing the tyrosine recombinase to determine the downstream element boundary. All alignments and quality filtering were performed with MUMmer4[138] (nucmer settings: -mum; delta-filter settings: -m -l 2000 -i 90) and manually inspected.

To establish phylogenies for *Starship* cargo genes, we performed pairwise alignments of predicted proteins for each genome using blastP v.2.12[139]. To ensure that phylogenies are not biased by missed gene annotations, we performed pairwise alignments of the predicted protein sequence against the chromosome-level assemblies and draft assemblies of the global collection using Exonerate v.2.70[140] with the parameter --model protein2genome -minintron 20 --maxintron 3000. We aligned sequences with MAFFT v. 7.310[141] using the --maxiterate 1000 –localpair options. Phylogenetic trees were built using RAxML v.8[142] with the parameters -m GTRGAMMA for nucleotide sequences and -m PROTGAMMAAUTO for protein sequences with 1000 bootstrap replicates.

## Visualizations and statistical analyses

All described statistical tests were performed using R[143]. Analyses of differences among genetic clusters were performed using ANOVAs with the *multcomp* R package v.1.4-25[144]. Heatmaps were generated with the *pheatmap* R package v.1.0.12[145]. Phylogenetic trees were plotted with the ggtree R package v3.8.0[146]. Synteny plots of the region were plotted with the *genoplotR* v.0.8.11[147] and *gggenomes* v.0.9.9[148] R packages. The correlation plot was generated using the *corrplot* R

package v.0.92[149]. Additional graphics were produced using the *ggplot2* R package[121]. To analyze associations of the RIP composite index and gene CNV events, we used a mixed-effect linear regression model with the R package *nlme* v.3.1.164[150]. We first used a baseline model of the explanatory variable (i.e., RIP mean per strain) and response variable (i.e., CNV event per strain) and compared it to more complex models adding population and RIP index as random effects. We used the ANOVA function from the package *car* v.3.1-2[151] to assess model fit. We used the function in r.squaredGLMM[152] to calculate the conditional ($R^2c$) and marginal coefficients ($R^2m$) of the generalized mixed-effect models. RIP composite data was retrieved from[42].

## Reporting summary

Further information on research design is available in the Nature Portfolio Reporting Summary linked to this article.

## Data availability

All sequencing data are available from the NCBI Sequence Read Archive. Individual accession numbers are reported in Supplementary Data 1. Supplementary Data 1, 2 and 5–11 are available in the Supplementary_Data_1-2_5-11.xlsx file. Supplementary Data 3 and 4 are available from https://zenodo.org/records/11616291[153].

## Code availability

Code used for analyses can be found in Zenodo (https://zenodo.org/records/8344848)[154]: scripts for plotting Figs. 1–5 can be found in the GATK_CNV_caller.zip and General.zip files, scripts for Fig. 6 can be found in the GEA.zip file, and scripts for Fig. 7 can be found in the swordfish.zip file within the repository[154].

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

## Acknowledgements
We would like to thank Thomas Badet for facilitating access to genome assembly data. We thank lab members for their thoughtful discussions and input. E.G.-T. was supported by funding from the European Union's Horizon 2020 research and innovation program under the Marie Skłodowska-Curie grant agreement (grant number 890630). D.C. was supported by the Swiss National Science Foundation grants 177052 and 205401.

## Author contributions
S.M.T. and D.C. conceived the study; S.M.T. and E.G.-T. performed analyses; A.F. provided datasets; D.C. provided funding and supervised the work; S.M.T. and D.C. wrote the manuscript with input from E.G.-T. and A.F.

## Competing interests
The authors declare no competing interests.
