## [Peer Review File · Nature Communications]

Copy number variation introduced by a massive mobile element underpins global thermal adaptation in a fungal wheat pathogenREVIEWER COMMENTS

Reviewer #1 (Remarks to the Author):

This manuscript investigates variation in copy number variation (CNV) driven by a massive mobile element in the fungal wheat pathogen *Zymoseptoria tritici*. The authors analyzed over 1109 genomes sequences for CNV. Strains were sampled from across the globe including the region of domestication and origins. The authors identified how CNVs contributed to chromosomal polymorphism and environmental range adaptation and identified an ~68 kb cargo-carrying Starship element. This work is substantial and very rigorous with the exception mentioned below. I question the process for calling CNV for the accessory chromosomes (see below). The reference listed for the methods did not include any information on this process. Since much of the MS hinges on this analysis I have to suggest rejection and encourage resubmission following a major revision. However, the CNV calls for the core chromosomes seems very appropriate and the data is compelling. The authors used PacBio long read sequencing for 16 strains to validate CNV calls. The CNV calls for the core genome seem very rigorous and well done. Similarly, it is not clear to me how high-impact SNV are called? I did not see a reference to this in the methods. I am not sure all figures are mentioned in the results. Maybe some can be moved to Supplements if not mentioned.

Specific comments:

72 remove italics from genomes.

445 How did you validate CNV calls? As described below?

481 In my mind this statement needs further revision: "Core chromosomes with more than 1.25 times the core coverage were defined as likely duplicated. Accessory chromosomes with less than three times the core chromosome mean coverage were considered absent from the strain." This makes no sense to me. If coverage > 1.25 means duplication, how come coverage <3 means absence for accessory chromosomes? Reference 44 cited to support this fact does not provide any information on media read depth used for accessory chromosomes. Figure 1B reflects this bias showing $(3-1.25)=1.75$ times reduction in CNV calls in accessory genomes. I am not convinced this is done well. Line 88: The validation of CNV does not avoid the bias mentioned above. Did the PCR validation of CNV for 14 genes involve accessory chromosomes?

I believe that the CNV calls for core chromosomes are solid, but not for accessory chromosomes. Figure 2A. I assume this show the average coverage for the populations of 1109 genomes? The legend states that "Expected read coverage was defined as a minimum of 1/3 read coverage compared to the mean core chromosome coverage." Why 1/3? If the accessory chromosome is present in a individual, why would you observe 1/3 the reads?

Figure 2F What are the numbers under total?

123 Figure 3F does not show lack of evidence for undescribed accessory chromosomes? I am lost here.

190 How were high-impact SNVs determined? I cannot find any methods regarding this

204 How were populations defined?

Figure 1D,E A candidate effector fixed in most of the populations including the center of domestication is of interest. Why is this locus released from selection in Africa? Very interesting pattern asking for functional validation.

519 Bonferroni correction is likely too stringent potentially inflating in false negatives. No action needed, just a comment.

525 All transcriptome data is already published.

567 Cargo genes? Is this accepted terminology or a starship pun?

Reviewer #2 (Remarks to the Author):

In this manuscript, the authors have used a population genomic dataset of over a thousand individuals of the fungal wheat pathogen *Zymoseptoria tritici* to study copy number variation (CNVs, e.g. larger deletions and duplications). They identify a number of CNVs that are associated with climate variables, among them an old Starship transposable element carrying the Sir2 gene, which is more common in colder climates and deletions of this locus is more common in warmer

regions.

This study does a good job surveying CNVs in *Z. tritici*. I only have some smaller comments when it comes to the analysis and results, but I do think the presentation can be improved upon. One bigger issue is the figures: they are often difficult to understand, especially because the figure legends are not very helpful. Often the legend just describes what type of plot is and they don't explain what is important or what the point of the figure is. Furthermore the colors that are used are often low contrast and it is unnecessarily difficult to tell them apart (I personally find it difficult to tell apart the green and blue-grey that are used in many figures). On top of this, the references to the figures in the text are often wrong, making it difficult to understand what is what.

I would therefore recommend that most figures are redone with a better color scheme that prioritizes contrast and readability, and also that more effort is put into making it easier for the reader to interpret the figures by describing them better in the figure legends or the main text.

When it comes to the main text, I have a few comments:

- I found it a bit difficult to keep track of when you were talking about raw CNVs and when you were talking about genes that were duplicated or missing.
- You describe and plot the size of the CNVs in bp, but it would perhaps also be interesting to see how many genes each CNV contain,.
- You have an earlier publication doing a similar analysis on a smaller dataset (Hartmann and Croll, 2017). I was surprised not to see you comparing your results here with the results from that study. I think it could be very interesting to do such a comparison and discuss to what extent they differ or confirm each other.

Finally, I also couldn't find all supplementary tables. It seems like only a few of them are available at Zenodo.

Minor comments:

123: Fig 3F: reference to wrong figure?

153: The numbers presented here are different than in Fig3A. Also, what is a fixed CNV? Do you mean that 73% of all genes do not show presence/absence polymorphisms, but, 3%, 5% and 17% are common, rare and singletons? In that case you need to rephrase this sentence and explain what you mean better.

161: Does this not contradict the previous statement (on line 157) that you inferred that deletions were the most common CNV?

164: Should say "Supplementary figure S4D" not just "Supplementary figure 4D"

198: Fig S6D doesn't exist. Should probably be S7B.

200: There is no Figure 4G, should probably be 3G.

213: Could the association between RIP and deletions be caused by poor mapping of highly RIPed regions, that show up "deletions"?

216: Add paragraph break?

264&265: Reference to Figure 4C is probably wrong.

280: "...the region segregates a large insertion affecting transcriptionally active genes..." A word is missing in this sentence.

402: Sentence broken up by unnecessary period

413: "fixation or loss" -> "fixation of loss"?

450: Weird sentence "...seven sets of independently sequenced sets..."

453: How did you perform CNV calling here? Using the same pipeline as described above? The only detail you give is 1kb windows.

476: Reference 119 has already been listed as reference 47.

482: Why does less than 3 times the mean coverage mean it is absent? That seems like a very high.

Figures:

1C: How do I interpret the "conflicting loci" part?

1E: Why do the duplications have so much lower CNQ scores?

1F: I'm not sure how to interpret this plot. Also, wouldn't a (paired) bar plot make more sense?

2F: Difficult to tell the colors apart, especially Sweden, Switzerland and USA

3A: This figure is not a Venn diagram as stated in the legend.

3E: "Partial deletions" only listed in legend and not shown in figure.

S1A: What does the Y-axis show in this plot? The fraction of each category that is found in both the mapped reads and the assemblies?

S4A: This is not a Venn diagram.

S4C: This figure does not appear to be mentioned in the text.

S11: Please explain the figure legend why you interpret this tree as showing that Sir2 is an ancient duplication etc.

Reviewer #3 (Remarks to the Author):

This paper from the Croll group examines copy number variation in the fungal plant pathogen *Zymoseptoria tritici* across 1109 globally distributed isolates. The authors ask an important question and use an interesting dataset to explore it. I found the associations between isolate environment and genotype particularly interesting, although I have some technical questions there. In sum, I thought the paper was interesting but found some of the claims too strong (see below). Likewise, I found the writing a bit unclear in places. While I am not a *Zymoseptoria* biologist, this really is my area of expertise and I still found portions of the paper difficult to follow. I have detailed a few examples below in specific comments. Some small increases in explanation of the data and justifications of the conclusions would help make the paper more clear and more readable.

The authors make 3 major claims in the abstract.

1. Strong purifying selection on most CNVs

It is not clear to me what the evidence for this claim is, aside from not seeing CNVs for 76% of core chromosome genes. If the argument is that there are so many more on the accessory chromosomes, how do you rule out that these accessory CNVs are positively selected and core chromosome CNVs are just neutral?

2. Genomic defense mechanisms likely accelerated gene loss

I found this supported.

3. CNVs affecting secondary metabolism facilitated local adaptation along climatic gradients

I think this is overstated. The authors really found correlations which I agree are highly suggestive. But there is no smoking gun in my mind without functional validation. I would be happier with this claim if the authors weakened the language ("likely facilitated" for example). I assume from the methods that the authors are accounting for population structure here as such that potential variants aren't just arising in a region with warmer weather and being associated by relatedness.

I found myself confused a few times by some of the claims. For example, on line 156-157, the authors say "Using parsimony, we infer that most CNV events are likely gene losses." However, there is no data pointed to here. The immediate preceding line says "Most CNV genes (85.7%) share an ortholog in at least one sister species (Supplementary Figure S4B)" but having an ortholog does not mean gene loss, so I am left confused. There are additional examples below.

Broadly, the blue and green colors chosen for deletion and duplication are far too similar for me. I would have had a much easier time reading this paper with more contrast between the colors in the plots.

Specific Questions/Comments:

Line 123: "We found no evidence supporting yet undescribed accessory chromosomes (Figure 3F)." Unclear to me how this cited figure relates to this statement. I'm unclear how this specific question was addressed without de novo assemblies.

Line 153: "We found that 3.3%, 5.2% and 17.5% of CNVs were common (>1% frequency), rare (<=1% frequency) and singletons, respectively, across the 1000-genome panel for a total of 8511 loci (Figure 3A). Most genes (73.9%) showed no CNVs and are likely under strong purifying selection.": The percentages in this pair of sentences add up to 100% but seem to describe two different things. Perhaps the first line is intended to be that 3.3% of genes have CNVs with >1% frequency, etc. As written, this instead sounds like genes with CNVs are split into those following categories, which should themselves sum to 100%.

Line 157: "Most genes on accessory chromosomes(90%; n= 203) exhibited CNVs compared to 24% (n = 2021) of core chromosome genes (Figure 3A)." Does this frequency account for whole chromosomal copy number variation in the accessory chromosomes? Ie., are 90% of genes present in various numbers regardless when that particular accessory chromosome is present? Similarly, are there multiple haplotypes of given accessory chromosomes segregating in the wild that are then reading as large sets of CNVs in this analysis?

Line 164: I think this should be pointing to supplemental figure 4C not 4B?

Line 173: "Taken together, our findings show that gene deletions can be readily purged, likely due to deleterious effects" I don't know this I find this supported by the data. Rather, it looks like many duplications are readily purged, as there are many more singleton duplications than shared ones- the duplications are forming at some rate but being purged before they reach high frequency. Perhaps the authors are saying that only the non-deleterious deletions are persisting long enough even to be singletons, but this needs a more clear explanation for me to get to the authors original point.

Line 190-192: "Genes segregating deletions harbored higher impact variants compared to conserved genes and genes with duplications." Is this a chicken or an egg? Deletion alleles will presumably pick up more deleterious variants over time (no purifying selection if the gene already doesn't work). But do the functional alleles also have a higher rate of high impact variants that potentially led to loss of selection to maintain instead?

Line 200: Should be figure 3G not 4G?

Line 220: It would be helpful to break 4E into individual panels that could be referenced specifically in this section.

Response to reviewers

Reviewer #1 (Remarks to the Author):

*This manuscript investigates variation in copy number variation (CNV) driven by a massive mobile element in the fungal wheat pathogen *Zymoseptoria tritici*. The authors analyzed over 1109 genomes sequences for CNV. Strains were sampled from across the globe including the region of domestication and origins. The authors identified how CNVs contributed to chromosomal polymorphism and environmental range adaptation and identified an ~68 kb cargo-carrying Starship element. This work is substantial and very rigorous with the exception mentioned below.*

I question the process for calling CNV for the accessory chromosomes (see below). The reference listed for the methods did not include any information on this process. Since much of the MS hinges on this analysis I have to suggest rejection and encourage resubmission following a major revision. However, the CNV calls for the core chromosomes seems very appropriate and the data is compelling. The authors used PacBio long read sequencing for 16 strains to validate CNV calls. The CNV calls for the core genome seem very rigorous and well done.

RESPONSE: Yes, we agree that the CNV calling at the accessory chromosome level was insufficiently described, and the validation was not documented clearly enough. We thoroughly revised the relevant text and added new analyses for clarity. We describe the changes in more details below.

Similarly, it is not clear to me how high-impact SNV are called? I did not see a reference to this in the methods.

RESPONSE: We assessed the most likely impact based on the predicted impact on protein coding regions and used the tool SNPeff. We focused on high-impact SNVs variants that have likely strongly deleterious effects, *i.e.* stop coding mutations. We clarify details of the procedure in the method section for single nucleotide variant calling.

I am not sure all figures are mentioned in the results. Maybe some can be moved to Supplements if not mentioned.

RESPONSE: Thank you for checking this. We fixed the issues with figure references and moved indeed some content to the Supplements.

Specific comments:

72 remove italics from genomes.

RESPONSE: Fixed.

445 *How did you validate CNV calls? As described below?*

RESPONSE: Yes, the validation steps can be found in the next section. We rephrased the sentence for clarity and improved the CNV validation sections.

481 *In my mind this statement needs further revision: "Core chromosomes with more than 1.25 times the core coverage were defined as likely duplicated. Accessory chromosomes with less than three times the core chromosome mean coverage were considered absent from the strain." This makes no sense to me. If coverage > 1.25 means duplication, how come coverage <3 means absence for accessory chromosomes? Reference 44 cited to support this fact does not provide any information on media read depth used for accessory chromosomes. Figure 1B reflects this bias showing (3-1.25)=1.75 times reduction in CNV calls in accessory genomes. I am not convinced this is done well. Line 88:*

RESPONSE: We agree that this lacked clarity and sufficient validation. We revised our CNV filtering steps in terms of expected coverage, added more information in the main text and figure legends, and included new analyses to improve chromosome copy number assessments. We agree that the expectation of a core chromosome full duplication is 2 times the normalized read coverage. However, there exist also large partial duplications covering nearly the entire chromosome, which would be detected as a close to 2 times normalized coverage (bioRxiv preprint 10.1101/2023.07.14.549097). For clarity, we now defined chromosome duplications as having at least 1.5 times coverage and clearly indicate that this category includes both full and substantial partial duplications. We also performed a manual inspection of read mapping patterns for cases showing substantial coverage variation and present the outcome in Supplementary Figure S2B.

As shown in Figure 2A, accessory chromosomes in particular show more substantial variation in read coverage compared to the more conserved core chromosomes. We found that setting a cutoff for at least 60% genes covered to be an efficient method to define chromosome presence/absence validation (Figure 2B, Supplementary Figure S2A). We validate now this threshold though more properly by including analyses of complete genome assemblies available for 8 isolates, which were also included in the CNV calling procedure. Contrasting evidence for full-length assembly of chromosomes and CNV calls showed that we scored 100% consistency for these 8 isolates (see Supplementary Figure S2C).

The validation of CNV does not avoid the bias mentioned above. Did the PCR validation of CNV for 14 genes involve accessory chromosomes? I believe that the CNV calls for core chromosomes are solid, but not for accessory chromosomes.

RESPONSE: Our original submission included only PCR validation for core chromosome genes. To remedy this shortcoming, we analyzed previously conducted PCR assays across all eight accessory chromosomes. The PCR data was originally published in PLOS Genetics in 2013 (10.1371/journal.pgen.1003567) and confirmed the overwhelming proportion of our chromosome CNV calls. The validation section of the Results was updated to reflect the new approaches. Together with the other improvements in documenting how we validated CNV calls and adjustment of thresholds, we believe that the CNV calling of accessory chromosomes is robust.

Figure 2A. I assume this show the average coverage for the populations of 1109 genomes? The legend states that "Expected read coverage was defined as a minimum of 1/3 read coverage compared to the mean core chromosome coverage." Why 1/3? If the accessory chromosome is present in a individual, why would you observe 1/3 the reads?

RESPONSE: Correct, the coverage corresponds to the 1104 genomes remaining after quality filters. We now added that information to the figure legend. As outlined above, we improved the filtering and validated chosen cut-offs with additional data. For clarity, we removed the expected coverage and now include a new figure (Figure 2B) to complement Figure 2A to show how the gene density cutoff was applied and validation was implemented.

Figure 2F What are the numbers under total?

RESPONSE: As this panel was not essential, we decided to remove the figure to reduce information density.

123 Figure 3F does not show lack of evidence for undescribed accessory chromosomes? I am lost here.

RESPONSE: This was a mistake in how we referred to this figure panel.

190 How were high-impact SNVs determined? I cannot find any methods regarding this

RESPONSE: We used the tool SNPeff to on the SNV callset of the same global set of strains to predict functional effects for each variant. High-impact SNVs were defined based on SNPeff classifications as causing likely the most severe effects on the gene function (*i.e.* start/stop codon changes). The method

section contains more details under the “Single nucleotide variant call and de novo genome assembly” header.

204 How were populations defined?

RESPONSE: The populations were defined as genetic clusters in a population genomic study of the same strains (Feurtey 2022). The genetic structure was defined by searching for a minimal number of genetic clusters representing adequately the diversity across all genomes. We added the technical details to the manuscript.

Figure 1D,E A candidate effector fixed in most of the populations including the center of domestication is of interest. Why is this locus released from selection in Africa? Very interesting pattern asking for functional validation.

RESPONSE: We agree this is an extremely interesting target for functional validation if adequate wheat panels could be established to investigate the source of selection. Establishing wheat panels historically used in North Africa would be an important element to test the hypothesis.

519 Bonferroni correction is likely too stringent potentially inflating in false negatives. No action needed, just a comment.

RESPONSE: We agree obviously that it is a stringent cutoff.

525 All transcriptome data is already published.

RESPONSE: Yes, all transcriptome data used in this study to characterize CNV locus was previously published. The method section “Transcriptome profiling” describes how the data was obtained originally and provides the relevant references.

567 Cargo genes? Is this accepted terminology or a starship pun?

RESPONSE: We followed the terminology coined by the original authors (Vogan et al 2021; Gluck-Thaler et al. 2022).

#####

Reviewer #2 (Remarks to the Author):

*In this manuscript, the authors have used a population genomic dataset of over a thousand individuals of the fungal wheat pathogen *Zymoseptoria tritici* to study copy number variation (CNVs, e.g. larger deletions and duplications). They identify a number of CNVs that are associated with climate variables,*

among them an old Starship transposable element carrying the Sir2 gene, which is more common in colder climates and deletions of this locus is more common in warmer regions.

*This study does a good job surveying CNVs in *Z. tritici*. I only have some smaller comments when it comes to the analysis and results, but I do think the presentation can be improved upon. One bigger issue is the figures: they are often difficult to understand, especially because the figure legends are not very helpful. Often the legend just describes what type of plot is and they don't explain what is important or what the point of the figure is.*

Furthermore the colors that are used are often low contrast and it is unnecessarily difficult to tell them apart (I personally find it difficult to tell apart the green and blue-grey that are used in many figures). On top of this, the references to the figures in the text are often wrong, making it difficult to understand what is what. I would therefore recommend that most figures are redone with a better color scheme that prioritizes contrast and readability, and also that more effort is put into making it easier for the reader to interpret the figures by describing them better in the figure legends or the main text.

RESPONSE: We thank the reviewer for these comments. We obviously agree that improvements are needed and changed the color scheme across all figures. We also added more information to the figure legends to improve clarity. We fixed the referenced figures in the text. More details are given below.

When it comes to the main text, I have a few comments:

- I found it a bit difficult to keep track of when you were talking about raw CNVs and when you were talking about genes that were duplicated or missing.

RESPONSE: We revised the first section of the Results to more adequately outline validation steps. We make it more clear at what point we switch to the “filtered” set of CNVs that we consider as reliable for all following analyses.

- You describe and plot the size of the CNVs in bp, but it would perhaps also be interesting to see how many genes each CNV contain,.

RESPONSE: Thank you for the suggestion. We now added this analysis as Supplementary Figure S5C and mention it in the Results section.

- You have an earlier publication doing a similar analysis on a smaller dataset (Hartmann and Croll, 2017). I was surprised not to see you comparing your results here with the results from that study. I think it could be very interesting to do such a comparison and discuss to what extent they differ or confirm each other.

RESPONSE: Yes, we referenced the study on a subset of about 10% of the currently available strains. We now discuss both results in the discussion section. Since the two manuscripts used quite different methods/validation steps beyond the PCR validation, a meaningful contrast in copy number variation is difficult. However, the earlier study is a subset of the study presented in this manuscript, we added elements to the discussion to compare the approaches.

Finally, I also couldn't find all supplementary tables. It seems like only a few of them are available at Zenodo.

RESPONSE: We regret this lack of clarity how to access the tables. The Supplementary Tables were separated into three files. File Supplementary_TablesS1-S2_S5-S11.xlsx contains Tables S1 to S11. Supplementary_TableS3.tsv contains Table S3 and Supplementary_table_S4.csv contains Table S4 due to their large size.

Minor comments:

123: Fig 3F: reference to wrong figure?

RESPONSE: Fixed.

153: The numbers presented here are different than in Fig3A. Also, what is a fixed CNV? Do you mean that 73% of all genes do not show presence/absence polymorphisms, but, 3%, 5% and 17% are common, rare and singletons? In that case you need to rephrase this sentence and explain what you mean better.

RESPONSE: The category "fixed" means gene with no CNV. We clarified the text. We also corrected the values and rephrased the sentence for clarity.

161: Does this not contradict the previous statement (on line 157) that you inferred that deletions were the most common CNV?

RESPONSE: Yes, this was poorly worded and adjusted the statement.

164: Should say "Supplementary figure S4D" not just "Supplementary figure 4D"

RESPONSE: Fixed

198: Fig S6D doesn't exist. Should probably be S7B.

RESPONSE: Fixed

200: There is no Figure 4G, should probably be 3G.

RESPONSE: Fixed

213: Could the association between RIP and deletions be caused by poor mapping of highly RIPed regions, that show up "deletions"?

RESPONSE: We agree that poor mapping could impact such associations in principle. However, in our case, the RIP composite index was based on read mapping to a TE consensus sequence library. Hence, the RIP index was not derived from any specific region with possible gene deletions. Furthermore, we expect RIP to stop introducing mutations at 10-20% divergence. At this level of divergence, we would not pick up even a highly RIPped sequence as a bona fide gene deletion passing our quality filters.

216: Add paragraph break?

RESPONSE: Break added.

264&265: Reference to Figure 4C is probably wrong.

RESPONSE: Reference fixed.

280: "...the region segregates a large insertion affecting transcriptionally active genes..." A word is missing in this sentence.

RESPONSE: We adjusted the sentence for clarity.

402: Sentence broken up by unnecessary period

RESPONSE: Fixed.

413: "fixation or loss" -> "fixation of loss"?

RESPONSE: We rephrased the sentence for clarity.

450: Weird sentence "...seven sets of independently sequenced sets..."

RESPONSE: We rephrased the sentence for clarity.

453: How did you perform CNV calling here? Using the same pipeline as described above? The only detail you give is 1kb windows.

RESPONSE: Yes, we used the approach described earlier in the method. We updated the text for clarity.

476: Reference 119 has already been listed as reference 47.

RESPONSE: Thank you, we fixed the reference list.

482: Why does less than 3 times the mean coverage mean it is absent? That seems like a very high.

RESPONSE: The thresholds were chosen to optimize false positive/negative calls. As outlined also in response to another reviewer, we have thoroughly revised how we present the CNV validation and have included additional controls. These include a validation based on completely assembled chromosomes for 8 strains also included in the study here. In addition, we integrated PCR validation data for accessory chromosomes. In conjunction, we believe that these thresholds are appropriate and hope that the revised text is clearer in outlining our approach.

Figures:

1C: *How do I interpret the “conflicting loci” part?*

RESPONSE: We meant to indicate genes with divergent CNV calls among sequencing replicates. We added more information to the figure legend for clarity.

1E: *Why do the duplications have so much lower CNQ scores?*

RESPONSE: CNQ score are defined as the difference between the two best genotype Phred-scaled log posteriors. Unless the gene duplication is very recent, reads mapping to duplicated regions will likely reveal genetic variability between the copies. We also think that the partial duplications are leading to higher uncertainty. Deletion loci are in principle more straightforward to call.

1F: *I'm not sure how to interpret this plot. Also, wouldn't a (paired) bar plot make more sense?*

RESPONSE: The plot shows the number of PCR validated genes that match the CNV calls. We now added more information to the figure legend for clarity.

2F: *Difficult to tell the colors apart, especially Sweden, Switzerland and USA*

RESPONSE: We revised color schemes throughout. We removed this particular panel due to a lack of relevance though.

3A: *This figure is not a Venn diagram as stated in the legend.*

RESPONSE: Legend adjusted.

3E: *"Partial deletions" only listed in legend and not shown in figure.*

RESPONSE: We adjusted the text to address this. This should have referenced 4D and 4E.

S1A: *What does the Y-axis show in this plot? The fraction of each category that is found in both the mapped reads and the assemblies?*

RESPONSE: This specific panel lacked relevance and we decided to remove it.

S4A: *This is not a Venn diagram.*

RESPONSE: Legend fixed.

S4C: *This figure does not appear to be mentioned in the text.*

RESPONSE: Added to the text.

S11: *Please explain the figure legend why you interpret this tree as showing that Sir2 is an ancient duplication etc.*

RESPONSE: We now added more information to the figure and to the figure legend.

#####

Reviewer #3 (Remarks to the Author):

*This paper from the Croll group examines copy number variation in the fungal plant pathogen *Zyoseptoria tritici* across 1109 globally distributed isolates. The authors ask an important question and*

use an interesting dataset to explore it. I found the associations between isolate environment and genotype particularly interesting, although I have some technical questions there. In sum, I thought the paper was interesting but found some of the claims too strong (see below). Likewise, I found the writing a bit unclear in places. While I am not a Zymoseptoria biologist, this really is my area of expertise and I still found portions of the paper difficult to follow. I have detailed a few examples below in specific comments. Some small increases in explanation of the data and justifications of the conclusions would help make the paper more clear and more readable.

RESPONSE: We are grateful for the very constructive evaluation. As detailed below, we improved the clarity of the writing and the figures. We also revised some statements to better reflect the evidence.

The authors make 3 major claims in the abstract.

1. Strong purifying selection on most CNVs

It is not clear to me what the evidence for this claim is, aside from not seeing CNVs for 76% of core chromosome genes. If the argument is that there are so many more on the accessory chromosomes, how do you rule out that these accessory CNVs are positively selected and core chromosome CNVs are just neutral?

RESPONSE: Yes, our statements required more supporting data and clarifications. We believe that the claim of purifying selection is most strongly supported by an excess of low frequency gene deletions compared to other, largely neutral types of polymorphisms (*i.e.* SNPs). We show now in the manuscript frequency spectra for these different types of polymorphisms and discuss the most likely interpretation. Hence, our claim does not rely on the difference in rates of gene CNVs on core vs. accessory chromosomes. As gene deletions seem to be pushed to lower population frequencies by purifying selection compared to nearly neutral polymorphism, the difference between accessory and chromosomes is most likely explained by relaxed selection on accessory chromosome genes. Such an interpretation is also supported by other types of analyses. E.g. accessory chromosome genes tend to be silenced under most conditions, segregate more point mutations and show higher rates of non-synonymous variants compared to core chromosome genes.

2. Genomic defense mechanisms likely accelerated gene loss

I found this supported.

3. CNVs affecting secondary metabolism facilitated local adaptation along climatic gradients

I think this is overstated. The authors really found correlations which I agree are highly suggestive. But there is no smoking gun in my mind without functional validation. I would be happier with this claim if the authors weakened the language ("likely facilitated" for example). I assume from the methods that the

authors are accounting for population structure here as such that potential variants aren't just arising in a region with warmer weather and being associated by relatedness.

RESPONSE: We are grateful for this point. The analyses were conducted in a way to correct for relatedness and, hence, unsupported associations with climatic variables. We agree completely though that association mapping alone is not sufficient to make strong claims about molecular mechanisms. We now include the term “facilitated” as suggested. We also clarified the method section Environmental and life-history trait adaptation analysis how relatedness was accounted for.

I found myself confused a few times by some of the claims. For example, on line 156-157, the authors say "Using parsimony, we infer that most CNV events are likely gene losses." However, there is no data pointed to here. The immediate preceding line says "Most CNV genes (85.7%) share an ortholog in at least one sister species (Supplementary Figure S4B)" but having an ortholog does not mean gene loss, so I am left confused. There are additional examples below.

RESPONSE: Thank you for this. Lines 156-157 were corrected. We meant that most deletions are likely true deletions based on the ortholog analyses among sister species.

Broadly, the blue and green colors chosen for deletion and duplication are far too similar for me. I would have had a much easier time reading this paper with more contrast between the colors in the plots.

RESPONSE: We agree that the color scheme was unhelpful. This is now thoroughly revised across all figures.

Specific Questions/Comments:

Line 123: "We found no evidence supporting yet undescribed accessory chromosomes (Figure 3F)." Unclear to me how this cited figure relates to this statement. I'm unclear how this specific question was addressed without de novo assemblies.

RESPONSE: We rephrased the sentence for clarity and fixed the figure reference.

Line 153: "We found that 3.3%, 5.2% and 17.5% of CNVs were common (>1% frequency), rare (<=1% frequency) and singletons, respectively, across the 1000-genome panel for a total of 8511 loci (Figure 3A). Most genes (73.9%) showed no CNVs and are likely under strong purifying selection.": The percentages in this pair of sentences add up to 100% but seem to describe two different things. Perhaps the first line is intended to be that 3.3% of genes have CNVs with >1% frequency, etc. As written, this

instead sounds like genes with CNVs are split into those following categories, which should themselves sum to 100%.

RESPONSE: Thank you. Rephrased for clarity now.

Line 157: "Most genes on accessory chromosomes(90%; n= 203) exhibited CNVs compared to 24% (n = 2021) of core chromosome genes (Figure 3A)." Does this frequency account for whole chromosomal copy number variation in the accessory chromosomes? I.e., are 90% of genes present in various numbers regardless when that particular accessory chromosome is present?

RESPONSE: Yes, we accounted for the fact that an accessory chromosome could be deleted altogether. All gene-level CNV frequency analyses were restricted to strains carrying the relevant chromosome. So, the gene CNV count are restricted to true gene deletions are not confounded by entirely missing chromosomes. We added this clarification to the figure 3A legend.

Similarly, are there multiple haplotypes of given accessory chromosomes segregating in the wild that are then reading as large sets of CNVs in this analysis?

RESPONSE: Accessory chromosomes do show numerous but mostly small segmental deletions. Hence, haplotype variation should be relatively small in field populations. There are clear exceptions to this for chromosome 14, which we describe in the text and Supplementary Figure S3. Chromosome 18 presented a high degree of CNVs classified as "complex" and such loci were removed from further gene CNV analyses.

Line 164: I think this should be pointing to supplemental figure 4C not 4B?

RESPONSE: Fixed

Line 173: "Taken together, our findings show that gene deletions can be readily purged, likely due to deleterious effects" I don't know this I find this supported by the data. Rather, it looks like many duplications are readily purged, as there are many more singleton duplications than shared ones- the duplications are forming at some rate but being purged before they reach high frequency. Perhaps the authors are saying that only the non-deleterious deletions are persisting long enough even to be singletons, but this needs a more clear explanation for me to get to the authors original point.

RESPONSE: Our statement lacked clarity. We do find indeed that deletions are removed by selection, and we show now supporting data on allele frequency spectra. Across the genome, deletion alleles

segregate at lower frequencies compared to nearly neutral SNP alleles, consistent with purifying selection acting against deletions.

Line 190-192: "Genes segregating deletions harbored higher impact variants compared to conserved genes and genes with duplications." Is this a chicken or an egg? Deletion alleles will presumably pick up more deleterious variants over time (no purifying selection if the gene already doesn't work). But do the functional alleles also have a higher rate of high impact variants that potentially led to loss of selection to maintain instead?

RESPONSE: We do indeed find that genes showing a segregating deletion in the species, tend to carry more high impact variants (in non-deletion alleles). So, this presents indeed a chicken-and-egg situation and it would require ancestral reconstruction of sequences adjacent to the deletion locus to identify whether high impact variants or deletion occurred first.

Line 200: Should be figure 3G not 4G?

RESPONSE: Fixed

Line 220: It would be helpful to break 4E into individual panels that could be referenced specifically in this section.

RESPONSE: Thank you. We modified the figure and adjusted the references

REVIEWERS' COMMENTS

Reviewer #2 (Remarks to the Author):

The authors have addressed most of my issues, but the supplementary figure legends are in many cases still very terse. I would recommend spending some effort improving these and making them more helpful to the reader. I have some specific requests below as well.

Minor comments:

Title: If you change the abstract and text to say that CNVs "likely" facilitated local adaptation, it seems weird to keep the old title.

93: "Figure S1AB" -> "Figure S1A-B"

138: "Strains with missing or duplicated chromosomes were removed from further analyses". Does this mean only strains with 21 chromosomes were included? Please state somewhere the total number of strains included in this analysis.

152: "The distinct frequency patterns of duplications and deletions were consistent across quality filtering stages (Supplementary Figure S4E)". Fig S4E shows a huge increase in rare CNVs, so this statement doesn't seem correct.

209: "S9AB" -> "S9A-B"

222: "We found that 85.7% of the associated genes share an ortholog in at least one *Zyoseptoria* sister species." You've already mentioned this earlier in the text, and it doesn't seem to fit very well in this context.

260: 64.5% + 14.3% does not add up to 100%. What is the rest?

537-543: Reference in wrong format, for instance "(Camacho 2009)" and two other cases.

Figures:

S2C: "The left panel x-axis refers to Illumina read coverage". The X-axis seems to show strain names?

S3A: Please explain in the legend how to interpret this figure. For instance, in the first panel/sample only chromosome 18 and 19 are present, in the second only 14 and 19, in the third none and in the fourth all?

S3B: I think this figure would be easier to parse as a paired bar plot (like for instance Fig 1D).

S3C: What does "single" mean?

S3D: Explain the panel with expression data better. What does "dpi" mean for instance?

S4B: Where is the secondary peak in the Oceania population? It looks pretty similar to the global population.

3B & 3D: Add legend to figure, or mention that the same legend applies to several figures.

S4D: The color in the plots and the colors in the legend do not match. Also, how can singletons have a density distribution? This figure needs a more detailed description to be parsed.

3D: The results presented here seems somewhat contradictory to the ones presented in Fig 3C. Just looking at 3C, I would say approximately 10-15% of all duplications are found in more than

one strains (ie are not singletons), but in 3D i would say that at least 20% seem to be found in more than population. How is this possible? Or am I just misreading the figures?

3E: Partial deletions are mentioned in the legend, but does not seem to appear in figure.

S5C: Why are the sample sizes (numbers on box plots) different in the top and bottom panels?

S9B: This figure is a bit small and there is space to make it bigger.

S10A: Explain what the size of the dots and triangles signify in this figure.

6B: Explain what each bar represents in the right panel.

S11: This tree is still hard to understand. In the text you talk about the gene of interest as Sir2, but here you seem to label is "GEA-CNV". You label two genes in *Ramularia* as "Sir5 paralogue", but it seems like the second one might actually be a Sir5 ORTHologue? Furthermore, there are several genes in other species called "SIR2 family" and "Sirtuin-2", which makes the tree very confusing. I would recommend modifying the figure by adding boxes clearly marking all Sir5 orthologs and all *Z. triticii* Sir2 orthologs (or mark them in some other way).

S12A: Explain what the X-axis shows in this figure.

S12B: Explain how the haplotypes are defined.

S16B: The B) label is missing in the legend.

Reviewer #3 (Remarks to the Author):

I was reviewer no. 3 in the previous round. I'm happy the reviewers chose to revise the manuscript and am mostly happy with the revised version. My one hang-up is the new figure S4B that is intended to demonstrate that deletions are left-shifted in frequency relative to SNPs. All three of these plots are on different scales and the SNPs are actually not phased to ancestral state like the deletions are (thus the distribution ends at frequencies of 0.5). I suspect the authors are correct, but the point is not apparent from this presentation of data. At minimum, a statistical test comparing the distributions should be conducted.

Reviewer #2 (Remarks to the Author):

The authors have addressed most of my issues, but the supplementary figure legends are in many cases still very terse. I would recommend spending some effort improving these and making them more helpful to the reader. I have some specific requests below as well.

Minor comments:

Title: If you change the abstract and text to say that CNVs "likely" facilitated local adaptation, it seems weird to keep the old title.

RESPONSE: Thank you. We believe that the title is coherent with the evidence presented.

93: "Figure S1AB" -> "Figure S1A-B"

RESPONSE: Fixed.

138: "Strains with missing or duplicated chromosomes were removed from further analyses". Does this mean only strains with 21 chromosomes were included? Please state somewhere the total number of strains included in this analysis.

RESPONSE: Here, we meant that we removed duplicated chromosomes for further analyses and kept only single-copy chromosomes. We rephrased the sentence for clarity.

152: "The distinct frequency patterns of duplications and deletions were consistent across quality filtering stages (Supplementary Figure S4E)". Fig S4E shows a huge increase in rare CNVs, so this statement doesn't seem correct.

RESPONSE: Thanks for highlighting this. We meant indeed low frequency CNVs, i.e. with $\leq 1\%$ frequency in the population, which includes singletons and rare CNVs.

209: "S9AB" -> "S9A-B"

RESPONSE: Fixed.

222: "We found that 85.7% of the associated genes share an ortholog in at least one Zymoseptoria sister species." You've already mentioned this earlier in the text, and it doesn't seem to fit very well in this context.

RESPONSE: We agree and removed this superfluous repetition.

260: 64.5% + 14.3% does not add up to 100%. What is the rest?

RESPONSE: We found different haplotypes linked to gene presence/absence/duplications for Swordfish (see figure S12B). The remainder % are constituted by additional haplotypes. We clarified the text.

537-543: Reference in wrong format, for instance "(Camacho 2009)" and two other cases.

RESPONSE: Fixed.

Figures:

S2C: "The left panel x-axis refers to Illumina read coverage". The X-axis seems to show strain names?

RESPONSE: Fixed.

S3A: Please explain in the legend how to interpret this figure. For instance, in the first panel/sample only chromosome 18 and 19 are present, in the second only 14 and 19, in the third none and in the fourth all?

RESPONSE: Each box represents the CNV chromosomes with matching RNAseq data (n=76). For example, we only found cases of chromosome duplications for 18 and 19 (first panel). We improved the figure and legend for clarity.

S3B: I think this figure would be easier to parse as a paired bar plot (like for instance Fig 1D).

RESPONSE: Replotted as suggested.

S3C: What does "single" mean?

RESPONSE: Meant to be single-copy. Clarified in the figure.

S3D: Explain the panel with expression data better. What does "dpi" mean for instance?

RESPONSE: Dpi means days post infection. Added the information to the figure legend.

S4B: Where is the secondary peak in the Oceania population? It looks pretty similar to the global population.

RESPONSE: The global population included OC explaining the pattern. For clarity, we split the representation to show the OC pattern more explicitly.

3B & 3D: Add legend to figure, or mention that the same legend applies to several figures.

RESPONSE: We added this information to the figure legend.

S4D: The color in the plots and the colors in the legend do not match. Also, how can singletons have a density distribution? This figure needs a more detailed description to be parsed.

RESPONSE: Thank you, we corrected the color legend and extended the description.

3D: The results presented here seems somewhat contradictory to the ones presented in Fig 3C. Just looking at 3C, I would say approximately 10-15% of all duplications are found in more than one strains (ie are not singletons), but in 3D i would say that at least 20% seem to be found in more than population. How is this possible? Or am I just misreading the figures?

RESPONSE: Thank you. Fig. 3C frequency refers to the total number of duplications found in the dataset. In fig. 3D, the frequency refers to total number of CNVs shared between population and from that, what percentage are duplications.

3E: Partial deletions are mentioned in the legend, but does not seem to appear in figure.

RESPONSE: We clarified the unified color legend with the requested information.

S5C: Why are the sample sizes (numbers on box plots) different in the top and bottom panels?

RESPONSE: Thank you for the comment, there was an indadvertent shift in the panel dataset. We updated the figures S5C and 3E accordingly.

S9B: This figure is a bit small and there is space to make it bigger.

RESPONSE: We increased the font size.

S10A: Explain what the size of the dots and triangles signify in this figure.

RESPONSE: These refer to p-value. We added the information to the legend.

6B: Explain what each bar represents in the right panel.

RESPONSE: We added this information to the figure legend.

S11: This tree is still hard to understand. In the text you talk about the gene of interest as Sir2, but here you seem to label is "GEA-CNV". You label two genes in *Ramularia* as "Sir5 paralogue", but it seems like the second one might actually be a Sir5 ORTHOlogue? Furthermore, there are several genes in other species called "SIR2 family" and "Sirtuin-2", which makes the tree very confusing. I would recommend modifying the figure by adding boxes clearly marking all Sir5 orthologs and all *Z. triticii* Sir2 orthologs (or mark them in some other way).

RESPONSE: Thank you for these comments. There remains some degree of uncertainty in the phylogenetic reconstruction and the panel should reflect this. We believe that the most parsimonious hypothesis is the one presented regarding the *Ramularia* homologs. We clarify our description of the presented homologs.

Regarding gene identifiers, we tried to remain consistent with accession identifiers in the databases to avoid confusion. But we agree that some renaming would produce a more parsimonious gene family tree. Also, Sirtuin genes were not named consistently across organisms (e.g. *Sir2* for *C. neoformans* and *Hst1* for *S. cerevisiae*). To improve clarity, we added *sirtuin* common names based on *Homo sapiens* (Sirtuin 1-7) across clades and highlighted the ancient duplication.

S12A: Explain what the X-axis shows in this figure.

RESPONSE: The x-axis refers to the genes found in the Starship region. Their order is based on the IPO323 reference genome.

S12B: Explain how the haplotypes are defined.

RESPONSE: The haplotypes are defined based on the identity and variation of the genes carried by the Starship. Information added.

S16B: The B) label is missing in the legend.

RESPONSE: Fixed.

Reviewer #3 (Remarks to the Author):

I was reviewer no. 3 in the previous round. I'm happy the reviewers chose to revise the manuscript and am mostly happy with the revised version. My one hang-up is the new figure S4B that is intended to demonstrate that deletions are left-shifted in frequency relative to SNPs. All three of these plots are on different scales and the SNPs are actually not phased to ancestral state like the deletions are (thus the distribution ends at frequencies of 0.5). I suspect the authors are correct, but the point is not apparent from this presentation of data. At minimum, a statistical test comparing the distributions should be conducted.

RESPONSE: Thank you. We adjusted the panels to improve cross-readability. We also binned the data to be able to perform a contingency table test on the frequency spectra. The Fisher test is included with the legend information.